# Lactation-associated macrophages exist in murine mammary tissue and human milk

Dilay Cansever[1,12,13], Ekaterina Petrova [1,13], Sinduya Krishnarajah [1], Caroline Mussak[1], Christina A. Welsh [1], Wiebke Mildenberger[1], Kevin Mulder[2,3,4], Victor Kreiner [1], Elsa Roussel[1], Sebastian A. Stifter [1], Myrto Andreadou[1], Pascale Zwicky[1], Nicole Puertas Jurado[1], Hubert Rehrauer [5], Ge Tan [5], Zhaoyuan Liu [6], Camille Blériot[2,7], Francesca Ronchi[8], Andrew J. Macpherson[8], Florent Ginhoux [2,3,6,9], Giancarlo Natalucci [10,11], Burkhard Becher [1] & Melanie Greter [1]✉

Macrophages are involved in immune defense, organogenesis and tissue homeostasis. Macrophages contribute to the different phases of mammary gland remodeling during development, pregnancy and involution postlactation. Less is known about the dynamics of mammary gland macrophages in the lactation stage. Here, we describe a macrophage population present during lactation in mice. By multiparameter flow cytometry and single-cell RNA sequencing, we identified a lactation-induced CD11c⁺CX3CR1⁺Dectin-1⁺ macrophage population (liMac) that was distinct from the two resident F4/80^hi and F4/80^lo macrophage subsets present pregestationally. LiMacs were predominantly monocyte-derived and expanded by proliferation in situ concomitant with nursing. LiMacs developed independently of IL-34, but required CSF-1 signaling and were partly microbiota-dependent. Locally, they resided adjacent to the basal cells of the alveoli and extravasated into the milk. We found several macrophage subsets in human milk that resembled liMacs. Collectively, these findings reveal the emergence of unique macrophages in the mammary gland and milk during lactation.

Macrophages from different tissues are heterogeneous in origin and phenotype, and carry out organ-specific functions governed by environmental signals[1]. Many tissue-resident macrophage (TRM) populations are derived from embryonic precursors and self-renew locally, while others are continuously replenished by monocytes[1,2]. In the case of the mammary gland in mice, macrophages support ductal development during puberty, epithelial cell remodeling during estrous cycling, alveolar expansion during pregnancy and postlactation involution[3–7]; whether macrophages in the mammary gland also contribute to lactation is not fully understood.

[1]Institute of Experimental Immunology, University of Zurich, Zurich, Switzerland. [2]Gustave Roussy Cancer Campus, Villejuif, France. [3]Institut National de la Santé et de la Recherche Médicale (INSERM) U1015, Equipe Labellisée—Ligue Nationale contre le Cancer, Villejuif, France. [4]Université Paris-Saclay, Ile-de-France, France. [5]Functional Genomics Center Zurich, ETH Zurich/University of Zurich, Zurich, Switzerland. [6]Shanghai Institute of Immunology, Shanghai JiaoTong University School of Medicine, Shanghai, China. [7]Institut Necker des Enfants Malades, CNRS, Paris, France. [8]University Clinic for Visceral Surgery and Medicine, Bern University Hospital, University of Bern, Bern, Switzerland. [9]Singapore Immunology Network (SIgN), Agency for Science, Technology and Research (A*STAR), Singapore, Singapore. [10]Larsson-Rosenquist Center for Neurodevelopment, Growth and Nutrition of the Newborn, Department of Neonatology, University Hospital Zurich, Zurich, Switzerland. [11]Newborn Research, Department of Neonatology, University Hospital Zurich, Zurich, Switzerland. [12]Present address: Roche, Basel, Switzerland. [13]These authors contributed equally: Dilay Cansever, Ekaterina Petrova. ✉e-mail: greter@immunology.uzh.ch

At birth, the infant's immune system is considered immature and relies partly on the maternal transfer of passive immunity for neonatal protection and immune regulation[8]. Historically, breast milk was thought to transmit immunity to the developing infant predominantly through its immunoglobulin content[9]; however, human milk also contains maternal leukocytes[10–13], which increase in number during infections[14,15].

While insight into maternal lymphocytes in milk is growing, we have limited understanding of the origin, nature and trafficking of maternal myeloid cells and the significance of their transfer to the suckling offspring. Here, we analyzed and profiled the mammary gland macrophages in mice and identified a specific macrophage population that arose during lactation. We defined its origin, location and cytokine dependency, and identified its putative counterparts in human breast milk.

## Results

### LiMacs accumulate in the murine mammary gland during lactation

We first investigated how tissue remodeling in the mammary gland during lactation impacts the myeloid compartment. Using flow cytometry, we characterized myeloid cells from lactating mammary glands of C57BL/6 (wild type) mice compared with nonlactating mammary glands from virgin/pregestational wild-type controls. We detected dendritic cells (DCs), Ly6C[hi] and Ly6C[lo] monocytes, neutrophils and eosinophils in both groups of mice, which slightly increased in numbers during lactation (Fig. 1a and Extended Data Fig. 1a–c). The two main resident macrophage populations (F4/80[hi] and F4/80[lo]) previously described in the nonlactating mammary gland[4,16] were detected in both virgin and lactating mice and their cell number remained relatively consistent throughout the lactation period (at days 1, 4, 6, 8, 12, 15 and 21 postpartum (pp)) (Fig. 1a–c). We further detected a population of CD11c[+]F4/80[+]CD64[+]MHCII[+]CX3CR1[+] macrophages that was present during lactation and was negative for CD11b, Lyve1, CD169, CD206 or CD38 (Fig. 1a–c and Extended Data Fig. 1a,c). These 'lactation-induced macrophages' (liMacs), which were referred to as ductal macrophages in previous studies[4,17], markedly increased in number by approximately tenfold between day 1 and day 12 pp and constituted the majority of all myeloid cells from day 4 pp onwards (Fig. 1b,c and Extended Data Fig. 1c). LiMac numbers decreased again postlactation (day 21 pp) and were almost absent in the mammary gland in virgin mice and rare during late gestation (E18.5) (Fig. 1a–c and Extended Data Fig. 1a–d).

To assess the spatial distribution of liMacs, immunohistochemistry of tissue sections from *Cx3cr1*[GFP/+] reporter mice were stained with CD11c antibodies, which allowed the identification of *Cx3cr1*-GFP[hi] CD11c[hi] liMacs. The virgin mammary gland contains a sparse ductal network, which expands during pregnancy to generate densely packed alveoli that produce milk during lactation[18]. The ducts and alveoli consist of an outer layer of SMA[+] (smooth muscle actin) basal cells and an inner layer of luminal cells. In *Cx3cr1*[GFP/+] virgin mammary glands, *Cx3cr1*-GFP[hi]CD11c[hi] liMacs were rare, although occasional cells were detected in nearby ducts (Fig. 1d). In contrast, *Cx3Cr1*-GFP[hi]CD11c[hi] cells were abundant in the lactating mammary gland and resided mostly inside or adjacent to the alveoli, wrapping the inner surface of the basal cell layer (Fig. 1d), in agreement with previous observations[4,17,19]. *Cx3cr1*-GFP[hi]CD11c[hi] liMacs were also MHCII[hi] (Extended Data Fig. 2a), consistent with the flow cytometry data (Fig. 1a). MHCII[hi] *Cx3cr1*-GFP[dim/–]CD11c[+/–] cells, likely representing F4/80[hi] and F4/80[lo] macrophages and Lyve1[+]*Cx3cr1*-GFP[dim/–]CD11c[dim] F4/80[hi] macrophages localized outside the alveoli (Extended Data Fig. 2a–b). cDCs are also CD11c[hi]MHCII[hi] and may express CX3CR1 (ref. 20). To rule out that *Cx3cr1*-GFP[hi]CD11c[hi]/MHCII[hi] cells represented cDCs, we stained lactating *Cx3cr1*[GFP/+] mammary glands with the common macrophage markers Iba1 and CD64. *Cx3cr1*-GFP[hi] cells were unanimously Iba1[+] and

CD64[+] (Extended Data Fig. 2c–d). As such, liMacs were detected from day 1 pp in the lactating mammary gland, increased during lactation until day 12 pp and were localized at the site of milk production within the mammary tissue.

### Murine liMacs exhibit a distinct transcriptome signature

To further characterize murine liMacs, we performed single-cell RNA-seq of sorted CD11b[+] and/or CD11c[+] myeloid cells from wild-type virgin and lactating mammary glands at day 7 pp. We identified nine distinct myeloid cell populations in the lactating mammary gland and eight myeloid cell populations in the virgin mammary gland (Fig. 2a and Extended Data Fig. 3a). In addition to the liMac (*Itgax*[+]*Cx3cr1*[+]) population, which was the most abundant population in the lactating mammary gland, we detected cDC1s (*Xcr1, Clec9a*), cDC2s (*Cd209a*), pDCs (*Ccr7*), Ly6C[hi] (*Lys2, Ly6c2, Ms4a4c*) and Ly6C[lo] (*Itgal, Nr4a1*) monocytes, neutrophils (*S100A8, S100A9, Clec4d*), F4/80[hi] and F4/80[lo] macrophages (Fig. 2a and Extended Data Fig. 3a). F4/80[hi] macrophages were classified as *Mrc1*[+]*Cd163*[+]*Lyve1*[+]*Folr2*[+]*Mgl2*[+]*Pf4*[+], genes reported to be expressed by TRMs associated with blood vessels (Extended Data Fig. 3b)[4,16,21–23]. F4/80[lo] macrophages were identified as *Ccr2*[+]*Ccl9*[+] *Fcrls*[+]*Lyz1*[+] (Extended Data Fig. 3a). LiMacs were CD11c[+] and MHCII[hi], similar to cDCs. However, in contrast to cDC1s and cDC2s, they did not express DC-lineage defining genes such as *Flt3* and *Zbtb46*, but highly expressed macrophage core genes including *C1qa, C1qb, Csf1r, Aif1* and *Fcgr1* (Extended Data Fig. 3c).

In the lactating mammary gland, 131, 155 and 103 genes were significantly differently expressed between liMacs, F4/80[hi] and F4/80[lo] macrophages, respectively (Fig. 2b). Highly expressed genes in liMacs included genes linked to the TGF-β signaling pathway (*Tgfbr1, Itgb5, Skil, Ntpcr* and *Lair1*) (Fig. 2c and Extended Data Fig. 3a). Among the top 100 genes expressed in liMacs, we identified genes such as *Tmem119* and *Hexb*, which have been described as microglia signature genes[24,25] (Fig. 2c). High expression of *Cadm1* and *Olfml3* was reported on tumor-associated TREM2[+] macrophages in the mammary gland[22]. While liMacs expressed *Cadm1* and *Olfm3* (Extended Data Fig. 3d), they were not specifically enriched for other signature genes described by these macrophages. Genes encoding several chemokines and cell adhesion molecules, including *Cxcl16, Glycam1, Itgav* and *Vcam1*, had also higher expression in liMacs compared with F4/80[hi] and F4/80[lo] macrophages (Fig. 2c). Accordingly, Gene Ontology analysis identified an enrichment in expression of genes associated with regulation of cell migration (Extended Data Fig. 3e).

Another gene highly expressed by liMacs was *Clec7a* (encoding Dectin-1) (Fig. 2c). Immunofluorescence staining of lactating mammary glands in *Cx3cr1*[GFP/+] mice showed that most GFP[+] cells were Dectin-1[+] (Extended Data Fig. 3f), suggesting Dectin-1 to be a useful marker for liMacs. LiMacs also expressed *Il1b* (Fig. 2c), in accordance with pathways upregulated in liMacs compared with F4/80[hi] and F4/80[lo] macrophages, which included cellular response to lipopolysaccharide (LPS), inflammatory response and phagocytosis (Extended Data Fig. 3e). On the other hand, F4/80[hi] macrophages were more associated with endocytosis pathways. LiMacs had higher expression of the matrix metalloproteinases *Mmp12* and *Mmp13*, which are involved in the regulation of tissue remodeling (Fig. 2c), and also expressed transcripts encoding main protein components of milk, such as *Csn2* (beta-casein), *Csn3* (kappa-casein) and *Wap* (whey acidic protein) (Fig. 2c and Extended Data Fig. 3a).

We also identified a small subset of actively proliferating liMacs (Extended Data Fig. 4a). To further assess whether the expression profile of liMacs changed over time, we performed scRNA-seq of sorted liMacs derived from lactating wild-type mice on day 8, 11 and 14 pp. Similar to liMacs at day 7 pp (Extended Data Fig. 4a), a small cluster of cycling liMacs, which expressed genes such as *Mki67* and *Top2a*, was detected at these three timepoints (Fig. 2d–e)[17]. The frequency of liMacs and proliferating liMacs did not change over time, and we did

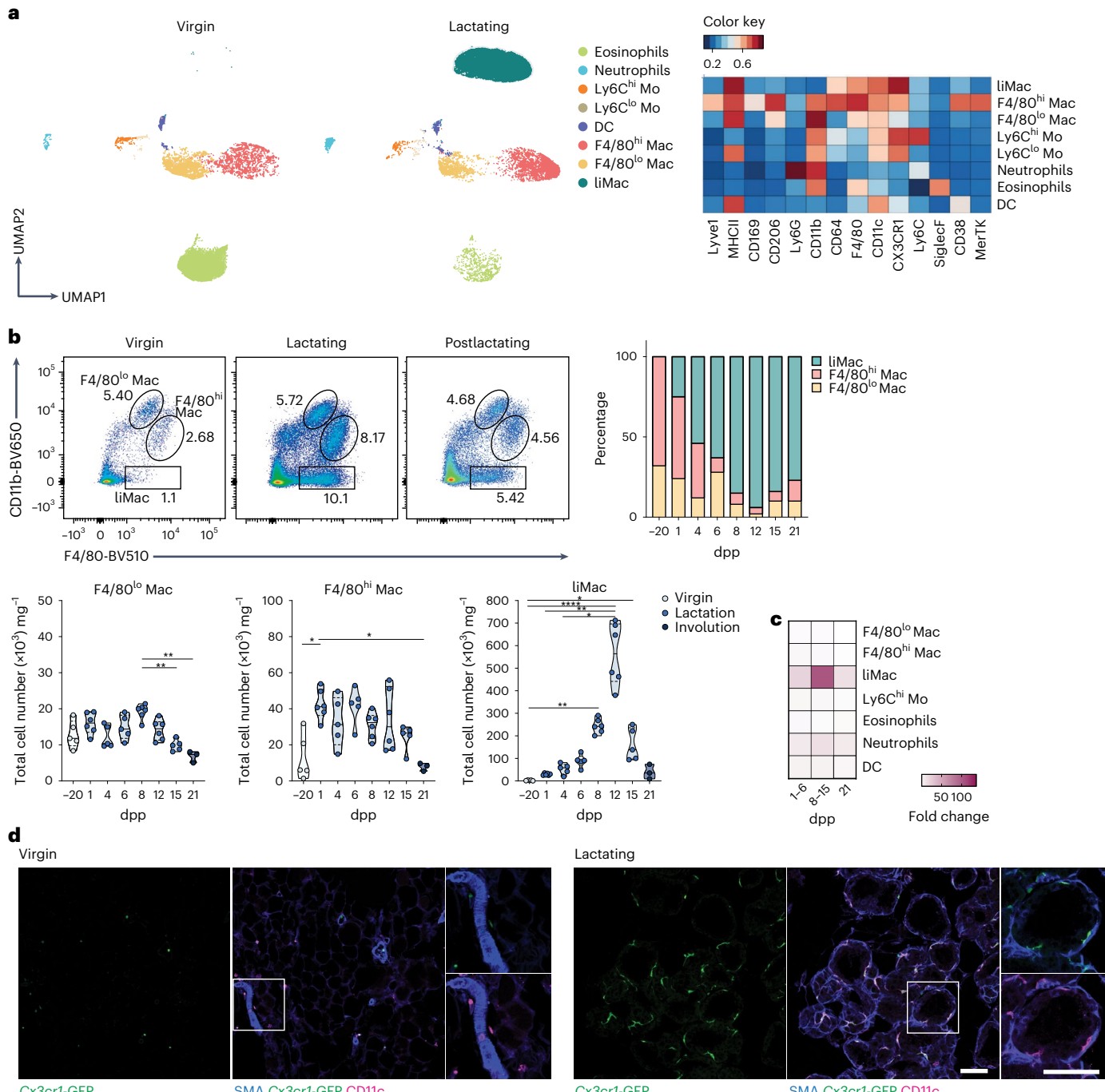

**Fig. 1 | Lactating murine mammary gland contains CD11c^hi macrophages.**
**a**,**b**, UMAP plots and corresponding heatmap (**a**) and representative flow cytometry plots, frequency and total cell counts (**b**) of the myeloid compartment (pregated on CD45⁺CD11b⁺ and/or CD11c⁺ cells) in the mammary glands of virgin or lactating (day 10 pp) (**a**) or virgin (day −20 pp), lactating (days 1–15 pp) or postlactating (day 21 pp) (**b**) wild-type mice, analyzed by flow cytometry. Heatmap shows the mean marker expression level. Data were transformed and percentile normalized, *n* = 3 per group (**a**). Data (*n* = 5–6 per timepoint) were pooled from six independent experiments (**b**). Kruskal–Wallis test was corrected

with Dunn's multiple comparisons test, *P < 0.05; **P < 0.01; ****P < 0.0001. Mac, macrophages, Mo, monocytes. **c**, Heatmap of the fold change in total cell counts of the different myeloid cell populations (as in **a**) in lactating mammary glands at early lactation (days 1–6 pp), late lactation (day 8–15 pp), postlactation (day 21 pp) compared with nonlactating mammary glands of wild-type mice. **d**, Immunohistochemistry of the virgin and lactating (day 8 pp) mammary glands from *Cx3cr1^GFP/+* mice. SMA (blue), *Cx3cr1*-GFP (green), CD11c (magenta). Insets, magnifications of outlined regions showing SMA, and CD11c or *Cx3cr1*-GFP. Images are representative of *n* = 2 mice. Scale bar, 75 μm.

not detect further temporal heterogeneity (Fig. 2d) or many differentially expressed genes comparing the three timepoints (Extended Data Fig. 4b). Altogether, the data indicated that the liMac population did not display plasticity during the lactating period in the healthy mammary gland.

**LiMacs derive from monocytes and expand during lactation**

As liMacs seemed to arise uniquely in the period between late pregnancy and early lactation, we tested whether they may be of monocytic origin. In *Ms4a3^CreR26^Ai14* mice, which allow the fate-mapping of monocytes and granulocytes[26], classical monocytes were efficiently

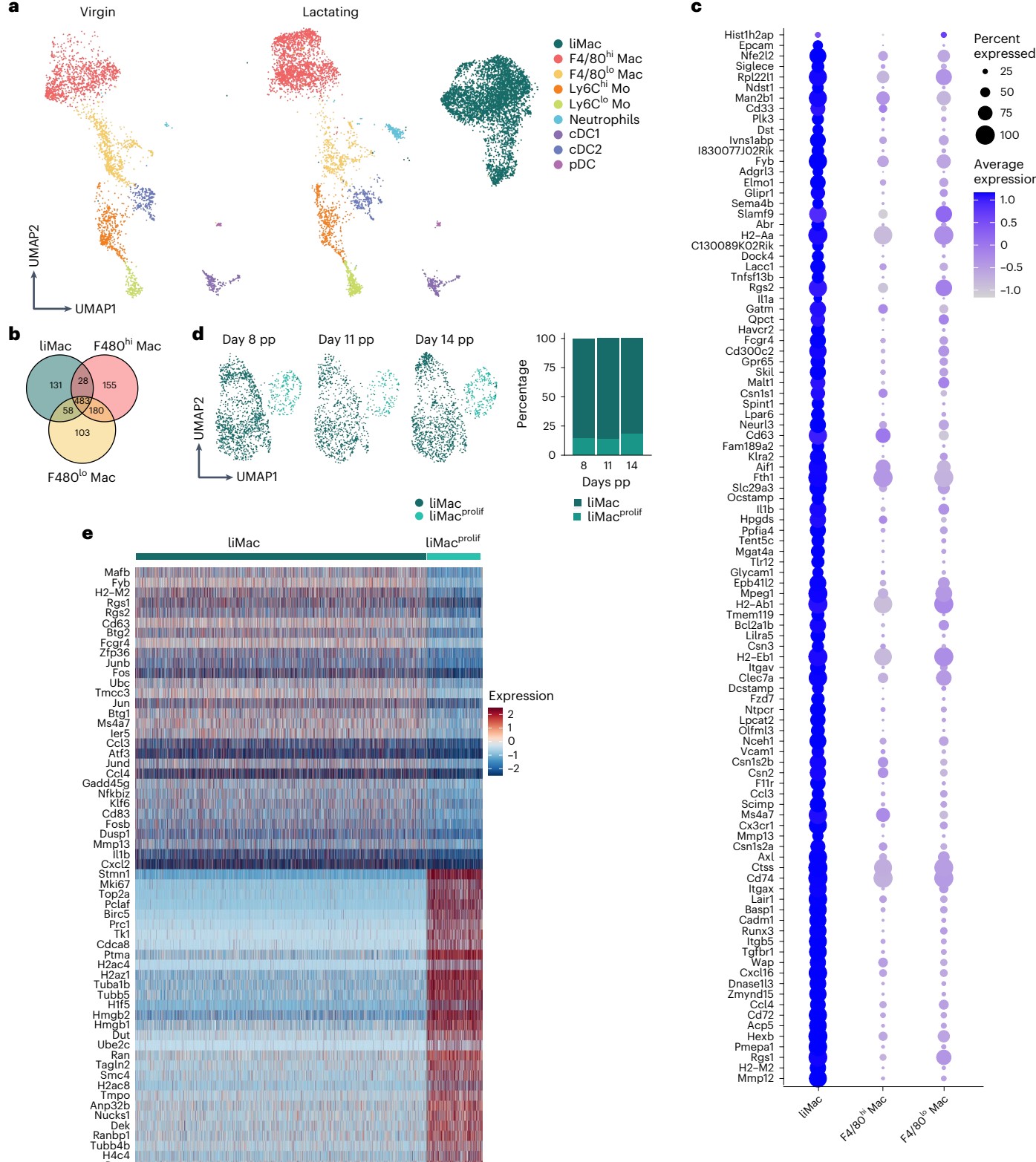

**Fig. 2 | Murine liMacs exhibit a unique transcriptional signature. a,b,** Seurat guided clustering and dimensionality reduction by UMAP showing eight and nine distinct myeloid cell populations (**a**) and Venn diagram of genes expressed in F4/80^lo macrophages, F4/80^hi macrophages and liMacs (**b**) in scRNA-seq performed on CD11b+ and/or CD11c+ myeloid cells sorted from virgin (*n* = 1) and lactating (day 7 pp, *n* = 2) mammary glands from wild-type mice. **c,** Dot plot of top 100 liMac signature genes with significantly higher expression in liMacs compared with the other defined immune populations, plotted versus F4/80^hi

and F4/80^lo macrophages (as in **a**). Dot size represents percentage of cells in a cluster expressing each gene; dot color reflects expression level (as indicated on legend). **d,e,** Seurat guided clustering and dimensionality reduction by UMAP showing liMac and proliferating liMacs (liMac^prolif) in scRNA-seq performed on sorted (CD45+Ly6G−Siglec-F−NK1.1−) CD11b+ and/or CD11c+ cells (**d**) and heatmap showing genes associated to liMac and liMac^prolif (**e**) in lactating mammary glands at day 8 (*n* = 3), day 11 (*n* = 3), and day 14 pp (*n* = 3) of wild-type mice.

labeled by tdTomato in the lactating mammary glands at day 10 pp (Fig. 3a). F4/80[hi] and F4/80[lo] macrophages are embryonically derived, but are slowly replaced by bone marrow (BM)-derived monocytes after puberty[4,16]. At day 10 pp, approximately 35% and 50% of F4/80[lo] and F4/80[hi] macrophages in the lactating mammary gland were labeled with tdTomato, respectively, along with 75% of the liMac population (Fig. 3a), suggesting that liMacs arise mostly from monocytes. Most Dectin-1[+] cells were tdTomato[+], as also shown using immunofluorescence (Fig. 3b).

To understand whether the rapid increase in liMac number during the first week of lactation was driven by proliferation or additional differentiation from monocytes, we used Ccr2[CreER]R26[Ai14] mice, in which tamoxifen administration leads to labeling of CCR2[+] cells, including monocytes and their progeny[27,28]. Ccr2[CreER]R26[Ai14] mice were treated with tamoxifen on days 1 and 3 pp and the lactating mammary glands were analyzed between days 8 and 15 pp. At these timepoints, only 10% of liMacs and F4/80[hi] macrophages were tdTomato[+], whereas nearly all monocytes were labeled with tdTomato (Fig. 3c). In addition, 85% of F4/80[lo] macrophages were tdTomato[+] (Fig. 3c), which was probably the result of direct labeling upon tamoxifen administration due to their expression of Ccr2 (Extended Data Fig. 3a). Using immunofluorescence, Iba1[+] cells that lined the alveoli were mostly negative for tdTomato in Ccr2[CreER]R26[Ai14] mice (Fig. 3d). These data suggested that the expansion of liMacs was independent of circulating monocytes during lactation. To confirm this, we used Ccr2[−/−] dams, in which the monocyte egress from the BM is blocked[29]. As anticipated, Ly6C[hi] monocytes were reduced drastically in the lactating Ccr2[−/−] mammary glands in comparison with wild-type mammary glands at days 13–15 pp (Fig. 3e). Also, numbers of F4/80[lo] macrophages were decreased markedly in lactating Ccr2[−/−] mammary glands, indicating their CCR2 dependence. In contrast, normal numbers of liMacs and F4/80[hi] macrophages were present in these mice (Fig. 3e), demonstrating that they accumulate independently of monocytes.

To further test that the liMacs expanded through local proliferation, we analyzed the proliferation rate of liMacs by administering EdU (5-ethynyl-2'-deoxyuridine) at day 7 pp. 20 hours post EdU administration, approximately 10% of liMacs, and only about 2% of F4/80[hi] or F4/80[lo] macrophages were EdU[+] (Fig. 3f). In agreement, immunohistochemistry of the lactating mammary glands from Cx3cr1[GFP/+] mice indicated that 13% of Cx3cr1-GFP[+] cells were Ki67[+] (Extended Data Fig. 4c). These data indicated that monocyte-derived liMacs expanded in situ during the course of lactation.

## CSF-1 and the gut microbiota modulate liMac development

The tyrosine kinase cell surface receptor CSF-1R is critical for the development and maintenance of most macrophage populations[30,31]. To test whether CSF-1R signaling also controlled liMac homeostasis, we generated Cd11c[Cre]Csf1r[fl/fl] mice, in which Csf1r was specifically deleted in CD11c[+] cells. LiMacs were almost completely absent in the lactating mammary glands in Cd11c[Cre]Csf1r[fl/fl] mice compared with Csf1r[fl/fl] mice (Fig. 4a), indicating a role for CSF-1R in the development of liMacs. F4/80[hi] and F4/80[lo] macrophages were not affected in Cd11c[Cre]Csf1r[fl/fl] mice (Fig. 4a), consistent with their lower expression of CD11c. The numbers of Ly6C[hi] monocytes, neutrophils and eosinophils were comparable in the lactating mammary glands of Csf1r[fl/fl] and Cd11c[Cre]Csf1r[fl/fl] mice (Extended Data Fig. 4d).

We then investigated the role of the two known CSF-1R ligands, the cytokines CSF-1 and interleukin-34 (IL-34)[32] in liMac development. The number of liMacs, F4/80[hi] and F4/80[lo] macrophages was similar in the lactating mammary glands of Il34-deficient (Il34[LacZ/LacZ]) mice[33] compared with Il34[+/+] or Il34[LacZ/+] mice at 7 days pp (Fig. 4b), indicating that IL-34 was dispensable for their generation. Given that Csf1[−/−] mice have developmental defects and die at a young age[30], wild-type dams were treated with CSF-1 neutralizing antibodies. This led to loss of liMacs in the lactating mammary gland and a reduction of around 25% and 50%

in F4/80[hi] and F4/80[lo] macrophages, respectively (Fig. 4c), whereas other cells, including Ly6C[hi] monocytes, were not affected (Extended Data Fig. 4e). Thus, liMacs were generated independently of IL-34 but required CSF-1-mediated CSF-1R signaling for their development and/or maintenance.

The maternal microbiota drives microbial colonization of the neonate, which is critical for the maturation of the immune system[34], and also influences macrophage development and function in the brain and gut[35–37]. To investigate its effect on mammary gland macrophages during lactation, we compared germ-free mice, mice colonized with four known bacteria (Escherichia coli MG1655, Bacteroides thetaiotaomicron ATCC29148, Lactobacillus reuteri I49 and Lachnoclostridium sp. YL32, referred to as 'Cuatro' mice) or conventionally housed wild-type mice. LiMacs were significantly less abundant in the lactating mammary glands of germ-free mice compared with wild-type or Cuatro mice, while the F4/80[hi] and F4/80[lo] macrophages remained largely unaffected (Fig. 4d). These data demonstrated that the generation of liMacs required CSF-1 and was partly influenced by the microbiome.

## LiMacs are detected in milk and exhibit immune functions

Given that Iba1[+] cells were occasionally detected in the lumen of the alveoli (Fig. 5a), we investigated whether liMacs extravasated into the milk. Milk was collected from lactating wild-type mice (at days 7–11 pp and 12–15 pp) and analyzed by flow cytometry. CD45[+] immune cells accounted for less than 1% of total milk cells including neutrophils (Ly6G[+]), eosinophils (Siglec-F[+]), Ly6C[hi] monocytes and CD64[+] cells (Fig. 5b,c). Within the CD64[+] cells, we detected a population of CD11b[−]F4/80[+]MHCII[+]CX3CR1[+]Dectin-1[+] macrophages (Fig. 5b), which correlated with the phenotype of liMacs, and slightly increased in frequency at days 12–15 pp of lactation (Fig. 5c). The equivalent of F4/80[hi] and F4/80[lo] mammary gland macrophages were not detected in the milk (Fig. 5b).

To interrogate whether liMacs were involved in remodeling of the mammary gland in lactation for milk production, we monitored the survival of neonates derived from wild-type mothers treated with CSF-1 antibodies (intraperitoneal (i.p.), every second day). Treatment with CSF-1 antibodies did not affect the survival or growth of pups (Fig. 5d). Equally, pups from Cd11c[Cre]Csf1r[fl/fl] dams also exhibited regular weight gain and general wellbeing (Fig. 5e), and the size and number of alveoli in the lactating mammary glands was comparable in Cd11c[Cre]Csf1r[fl/fl] and Csf1r[fl/fl] dams (Extended Data Fig. 5a). These observations suggested that liMacs did not play a critical role in tissue transformation for milk production.

Given that liMacs expressed genes encoding milk proteins (for example, Csn3), we explored whether they modulated the composition of milk directly or indirectly. Proteomics analysis of milk derived from Cd11c[Cre]Csf1r[fl/fl] or Csf1r[fl/fl] mice at day 7 pp indicated no important alterations in the 50 most abundantly expressed proteins between the two groups, including beta-casein (CASB), kappa-casein (CASK), lactalbumin (LALBA) or lactotransferrin (TRFL) (Extended Data Fig. 5b). The concentration of immunoglobulins did also not significantly change between milk derived from Cd11c[Cre]Csf1r[fl/fl] or Csf1r[fl/fl] mice (Extended Data Fig. 5c). In agreement, a similar number of IgA-producing plasma cells was present in the lactating mammary gland of CSF-1 antibody-treated and isotype control or untreated mice (Extended Data Fig. 5d,e). Thus, liMacs did not markedly affect the content of milk proteins, including immunoglobulins.

Pathway analysis of the scRNA-seq data indicated that liMacs highly expressed genes associated to phagocytosis, inflammatory responses and response to LPS, suggestive of a microbicidal function. We probed the capacity of liMacs to respond to Toll-like receptor (TLR) stimulation in vitro by exposing cells isolated from the lactating mammary gland to different TLR ligands, including zymosan (a ligand for TLR2) and LPS (TLR4). LiMacs had the highest expression of pro-IL1β in all conditions compared with F4/80[hi] and F4/80[lo] macrophages. Zymosan and

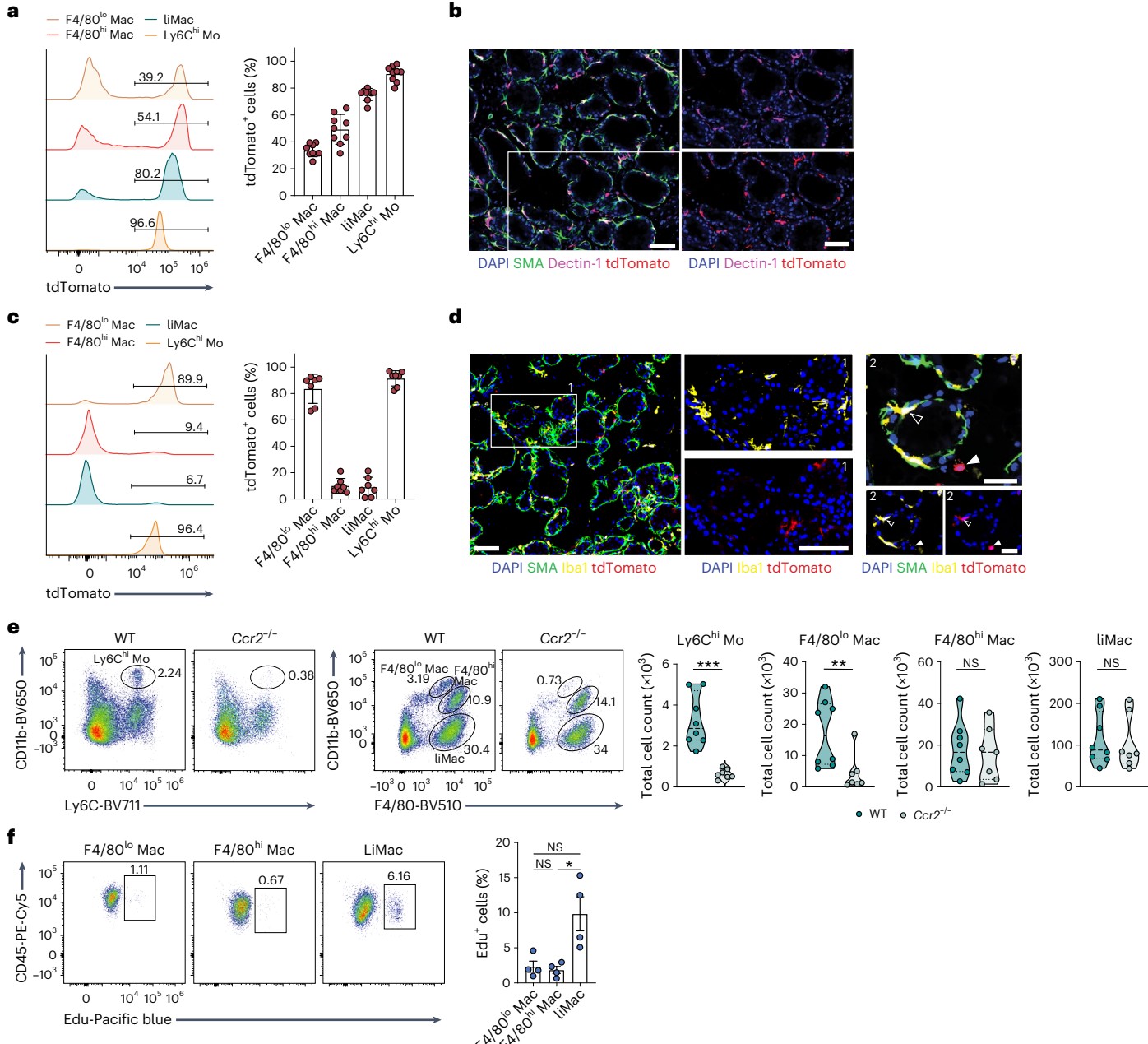

**Fig. 3 | Murine liMacs are derived predominantly from monocytes and expand during lactation. a**, Representative flow cytometry histograms and graph (±s.d.) showing the percentage of tdTomato labeling in F4/80$^{lo}$ macrophages (F4/80$^{lo}$ Mac), F4/80$^{hi}$ macrophages (F4/80$^{hi}$ Mac), liMacs (pregated on CD45$^{+}$SiglecF$^{-}$Ly6C$^{-}$Ly6G$^{-}$) and Ly6C$^{hi}$ monocytes (Mo) (pregated on CD45$^{+}$SiglecF$^{-}$Ly6G$^{-}$ cells) in mammary glands from lactating (day 10 pp) *Ms4a3$^{CreER}$R26$^{Ai14}$* mice. Pooled data from three independent experiments, *n* = 9. **b**, Immunohistochemistry of lactating mammary glands from *Ms4a3$^{CreER}$R26$^{Ai14}$* mice (day 10 pp). DAPI (blue), SMA (green), Dectin-1 (magenta), tdTomato (red). Right panels show DAPI and single stains for Dectin-1 and tdTomato, respectively, of the outlined region in the overview image on the left. Scale bar, 70 μm; *n* = 4. **c**, Representative flow cytometry histograms and graph (±s.d.) showing the percentage of tdTomato labeling in F4/80$^{lo}$ Macs, F4/80$^{hi}$ Macs, liMacs and Ly6C$^{hi}$ Mo as in **a** in the mammary glands of lactating (day 8–15 pp) *Ccr2$^{CreER}$R26$^{Ai14}$* mice treated with tamoxifen on day 1 and day 3 pp. Pooled data of three experiments, *n* = 6 (*n* = 4 from 8 days pp and *n* = 2 from 15 days pp). **d**, Representative

immunofluorescence image of lactating mammary glands (day 15 pp) from *Ccr2$^{CreER}$R26$^{Ai14}$* dams treated with tamoxifen on day 1 and day 3 pp. DAPI (blue), SMA (green), Iba1 (yellow) and tdTomato (red). Middle panels (1) and bottom panels right (2) show DAPI and single stains for Iba1 and tdTomato of the outlined region in the overview image on the left (1) or of the image on top (2), respectively. Scale bars, 70 μm for (1), 35 μm for (2); *n* = 6. **e**, Representative flow cytometry plots and violin plots show the frequency and total cell counts of monocytes, F480$^{hi}$ Macs, F4/80$^{lo}$ Macs and liMacs as in **a** in the mammary glands from lactating (days 13–15 pp) *Ccr2$^{-/-}$* and wild-type dams. Pooled data from five independent experiments; *n* = 8 (WT) and 7 (*Ccr2$^{-/-}$*). Two-tailed Mann–Whitney test was performed, **$P < 0.01$; ***$P < 0.001$; NS, not significant. **f**, Representative flow cytometry plots and a graph (±s.e.m.) showing the percentage of EdU$^{+}$ F4/80$^{lo}$ macrophages, F4/80$^{hi}$ macrophages and liMacs in mammary glands from lactating (day 7 pp) wild-type mice 20 h post EdU injection. Pooled data from three independent experiments, *n* = 4. Kruskal–Wallis test followed by Dunn's multiple comparison test. *$P < 0.05$.

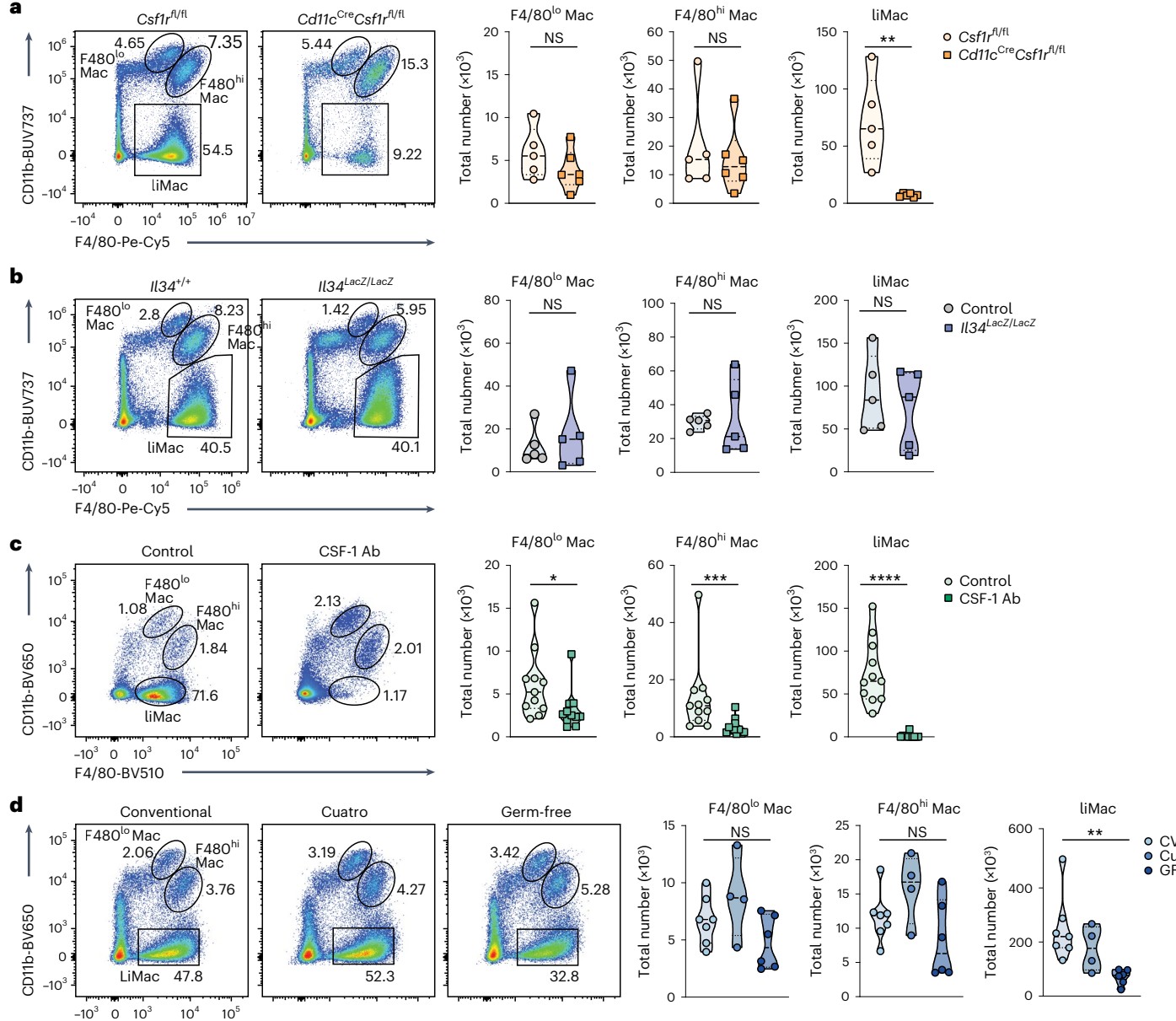

**Fig. 4 | CSF-1 and the microbiota regulate the development of murine liMacs in the lactating mammary gland. a**, Representative flow cytometry plots and violin plots showing the frequency and total cell counts of F4/80[hi] macrophages (Mac), F4/80[lo] Macs and liMacs in lactating mammary glands (days 8–14 pp) from control (WT or *Csf1r*[fl/fl]) and *Cd11c*[Cre]*Csf1r*[fl/fl] dams. Pooled data from five to six mice analyzed in four independent experiments. Two-tailed Mann–Whitney test was performed. **P* < 0.01. **b**, Representative flow cytometry plots and violin plots showing the frequency and total cell counts of F4/80[hi] Macs, F4/80[lo] Macs and liMacs (per 10[6] total cells) in lactating mammary glands (day 9 pp) from *Il34*[LacZ/LacZ] and control (*Il34*[+/+] or *Il34*[LacZ/+]) dams. Pooled data of five mice analyzed in four independent experiments. Two-tailed Mann–Whitney test was performed. **c**, Representative flow cytometry plots and violin plots showing the frequency and total cell counts of F4/80[hi] Macs, F4/80[lo] Macs and liMacs in lactating mammary glands (days 7–14 pp) from dams treated with CSF-1 antibodies (Ab) (two to three times) or control dams (treated with isotype antibody or left untreated). Pooled data from 11 mice analyzed in five independent experiments. Two-tailed Mann–Whitney test was performed. **P* < 0.05, ****P* < 0.001, *****P* < 0.0001. **d**, Representative flow cytometry plots and violin plots showing the frequency and total cell counts of F4/80[hi] Macs, F4/80[lo] Macs and liMacs in lactating mammary glands (day 7 pp) of conventional (CV), Cuatro (Cu) and germ-free (GF) dams. Pooled data from three independent experiments are shown; *n* = 4–7. Kruskal–Wallis test followed by Dunn's multiple comparison test. **P* < 0.01.

LPS-treated liMacs displayed increased production of tumor necrosis factor (TNF) compared with untreated liMacs, but this was consistently lower than in F4/80[hi] macrophages (Extended Data Fig. 6a,b).

To assess the phagocytic capability of liMacs, lactating mammary gland macrophages were exposed in vitro to pH-sensitive pHrodo Red zymosan, *E. coli* or *S. aureus* bioparticles. *S. aureus* and *E. coli* are frequently associated with mastitis. LiMacs phagocytosis of *E. coli* particles was higher than that of F4/80[hi] and F4/80[lo] macrophages, whereas F4/80[lo] macrophages displayed increased internalization of zymosan particles (Extended Data Fig. 6c). No significant difference in particle uptake was found between liMacs, F4/80[hi] and F4/80[lo] macrophages upon exposure to *S. aureus* particles (Extended Data Fig. 6c). These data indicated that liMacs were phagocytic and responsive to microbial stimuli.

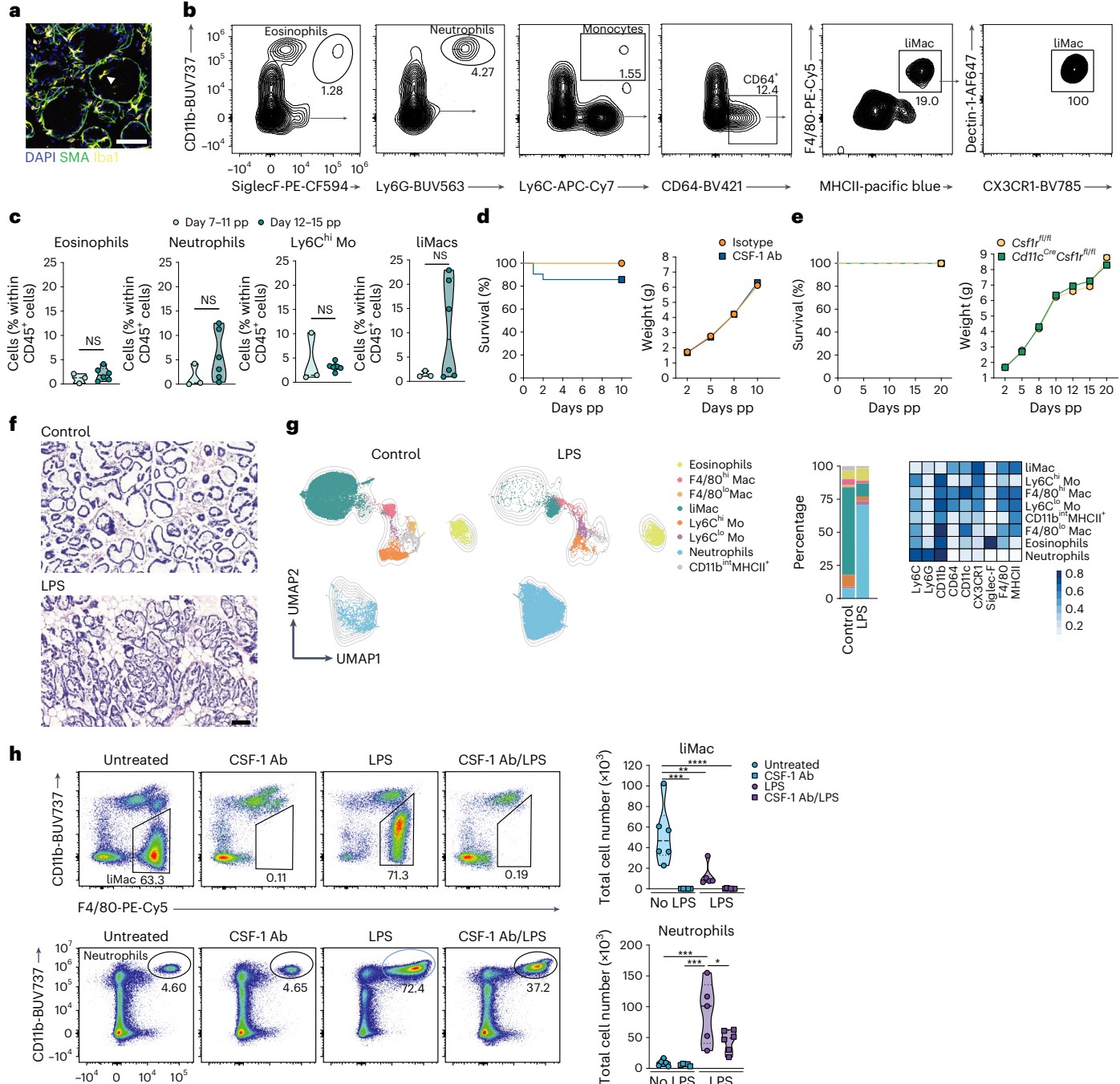

**Fig. 5 | LiMacs are present in murine milk and contribute to neutrophil recruitment in mastitis. a**, Representative immunofluorescence image of a lactating mammary gland (day 15 pp). DAPI (blue), SMA (green) and Iba1 (yellow). Arrowhead shows an Iba1+ cell inside an alveoli. Scale bar, 70 μm. **b,c**, Representative flow cytometry plots and gating strategy (**b**) and violin plots showing frequencies of eosinophils, neutrophils, Ly6Chi monocytes and liMacs within CD45+ cells (**c**) in lactating mammary glands (days 7–11 and 12–15 pp) from wild-type mice; *n* = 3 for days 7–11 pp, and *n* = 6 for days 12–15 pp, data pooled from two independent experiments. Two-tailed Mann–Whitney test was performed. **d**, Survival curve and weight of WT pups nursed by WT dams treated with CSF-1 antibodies or isotype control at E18.5 and every other day pp; *n* = 15, pooled data from three independent experiments. **e**, Survival curve and weight of pups (*Cd11c*Cre*Csf1r*fl/+ or *Csf1r*fl/+) nursed by *Cd11c*Cre*Csf1r*fl/fl or *Csf1r*fl/fl mice; *n* = 30, pooled data from four experiments. **f**, H&E staining of mammary gland sections from unchallenged (control) WT mice or WT mice injected with LPS into the mammary gland (18 h post LPS challenge). Scale bar, 100 μm;

*n* = 2–3 mice per group. **g**, UMAP plots and bar graph showing the frequencies of eosinophils, F4/80hi macrophages (Mac), F4/80lo macrophages, liMacs, Ly6Chi monocytes (Mo), Ly6Clo monocytes, neutrophils and CD11bintMHCII+ cells (pregated on CD45+CD3−CD19−NK1.1− and CD11b+ and/or CD11c+ and/or F4/80+ cells) in lactating mammary glands (days 11–14 pp) of WT mice, injected with LPS into the fourth mammary glands 18 h earlier or left untreated (control). The corresponding heatmap shows the median marker expression level. Data were transformed and quantile normalized, *n* = 2–3 per group. **h**, Representative flow cytometry plots and violin plots showing total cell counts of liMacs (top) (pregated on CD45+CD19−NK1.1−CD3−SiglecF−Ly6G−) and neutrophils (bottom) (pregated on CD45+LiMacs−) in lactating mammary glands (days 10–14 pp) in WT mice left untreated or pretreated with CSF-1 antibodies on days 7–11 pp, and injected with LPS into the fourth mammary glands 18 h earlier or left uninjected. Data (*n* = 5–6 per group) were pooled from two independent experiments. One-way ANOVA test was applied, *P < 0.05, **P < 0.01, ***P < 0.001, ****P < 0.0001.

To further interrogate the potential role of liMacs in the inflamed mammary gland in vivo, we used a model of acute experimental mastitis induced by LPS. LPS injection into the lactating mammary gland led to infiltration of immune cells into the alveolar spaces 18 h post LPS challenge, as shown by hematoxylin and eosin (H&E) staining (Fig. 5f). By flow cytometry of the LPS-challenged mammary gland, we found that, within myeloid cells specifically Ly6G[+] neutrophils were increased in frequency and total cell number (Fig. 5g,h), a hallmark of *E. coli*-induced mastitis[38,39]. Conversely, liMacs upregulated CD11b and their frequency and numbers were significantly decreased (Fig. 5g–h)—a phenomenon frequently reported for TRMs during inflammation[40]. Other mononuclear phagocytes (CD11b[+] and/or MHCII[+]) were also numerically decreased upon LPS challenge (Extended Data Fig. 6d). Depletion of liMacs by CSF-1 antibody treatment before LPS injection led to a significant reduction in neutrophil infiltration compared with untreated LPS-injected mice (Fig. 5h), suggesting a role of liMacs in the early inflammatory response, as previously proposed[41]. Altogether, while liMacs were dispensable for tissue remodeling for milk production, they trafficked into the milk, were equipped to respond to microbial stimulation and were implicated in the recruitment of innate immune cells upon a bacterial challenge.

## Human milk contains liMac-like macrophages

Mononuclear phagocytes in human milk have been profiled by flow cytometry and cells including macrophages in human breast milk by scRNA-seq[12,13]. We investigated whether they might be counterparts of murine liMacs and assessed the cellular immune compartment of human milk phenotypically and transcriptionally in samples from healthy donors collected within 4–63 days of delivery. Similar to murine milk, maternal leukocytes in human milk represented between 1% and 10% of total milk cells, which consist primarily of lactocytes. Lymphocytes accounted for up to 60% of CD45[+] immune cells (Fig. 6a,b). The remaining cells were myeloid cells, consisting of CD11b[+]CD66b[+]CD16[+] neutrophils, CD11b[+]Siglec8[+] eosinophils and CD11b[+]CD11c[+]CD64[+]HLA-DR[+]CD163[+]CD14[+]CD16[+] macrophages (Fig. 6a and Extended Data Fig. 7a).

scRNA-seq of leukocytes in human milk identified T cells (*CD3d, CD3e, CD247, TRAC*), B cells (*CD19, CD79b, BANK1*), DC2/DC3s (*CD1C, CD1E, CLEC10A*), monocytes (*FCN1, VCAN, S100A8, S100A9*) and macrophages (*MARCO, CD68, CD163, SIGLEC1, AIF1*) in all samples (Fig. 6c,d and Extended Data Fig. 7b). Among the macrophages, three clusters were detected. Cluster 1 (Mac 1) was enriched for expression of *APOE, SPI1* and *CD9*, whereas cluster 2 (Mac 2) and cluster 3 (Mac 3) expressed higher levels of *ITGAX, CLEC7A* and *CD163*, respectively (Fig. 6d). Mac 3 and Mac 1 may partially relate to the FOLR2[+] macrophages described in the healthy human mammary gland and in breast cancer (Extended Data Fig. 7c)[22]. Comparison of the liMac transcriptome with human milk macrophages indicated that many of the liMac signature genes, including *HEXB, CXCL16, CLEC7A, LPCAT2* or *MS4A7*, were also expressed in one or all three subsets of human macrophages (Fig. 6e), suggesting a partially conserved lactation-associated macrophage profile across species. Similarity score analysis between the murine lactating mammary gland macrophages and the three human milk macrophage subsets indicated that liMacs and murine F4/80[lo] macrophages resembled primarily the Mac 2 subset of human milk macrophages, while murine F4/80[hi] macrophages aligned to both Mac 1 and Mac 2 (Fig. 6f). This was confirmed by principal component (PC) analysis (PCA) of the three murine mammary gland and the three human milk macrophages (Extended Data Fig. 7d).

Next, we mapped the human milk macrophages onto a human scRNA-seq compendium across 13 human tissues from healthy and pathological states[42]. This analysis indicated that most human milk macrophages correlated to a macrophage cluster called 'TREM2_Mac' (no. 3) in the human scRNA-seq compendium[42] (Fig. 6g,h), which was characterized by the expression of *TREM2, APOE, APOC1, SPP1, CD9, LIPA, CD63* and *LGALS3* (Fig. 6i) and was specifically enriched

in macrophages from different tumors including breast cancer[22,42]. TREM2[+] macrophages with immunosuppressive properties have also been described in preclinical tumor models[43,44], and a similar TREM2 signature has been associated with lipid metabolism and phagocytosis[45]—a feature that may also correlate with human milk macrophages, which had pathways enriched for 'lipid metabolic process' and 'phagocytosis, engulfment' (Extended Data Fig. 7e). The human milk Mac 3 population corresponded partially to the 'macrophage population' (no. 11) in the human scRNA-seq compendium[42], which expressed metallothionein genes that were reported to have immunoregulatory properties[46]. These data indicated that macrophages in the human milk partly resembled murine liMacs in phenotype and transcriptome, and displayed a profile reminiscent of lipid- or cancer-associated macrophages.

## Discussion

Here we analyzed the macrophage compartment in the murine mammary gland and found, in addition to the F4/80[hi] and F4/80[lo] macrophage populations resident in the murine pregestational mammary gland, a population of CD11b[−]CD11c[hi]CX3CR1[hi]Dectin-1[+]MHCII[−] liMacs that emerged and accumulated in the lactating mammary gland[4,16,17]. LiMacs accounted for most mammary gland macrophages during lactation, were dependent on CSF-1 signaling and localized close to the mammary duct epithelium.

The F4/80[hi] macrophages in the mammary gland resembled a population of macrophages associated with blood vessels[21,22], while the F4/80[lo] population was more vague in terms of phenotype, profile, localization and lineage, and may be heterogenous. Whether monocytes during pregnancy contribute to the increased pool of F4/80[hi] macrophages observed early postpartum in the mammary gland remains to be shown. LiMacs originated mostly from BM-derived monocytes and expanded in situ rapidly during lactation, independently of CCR2. Apart from their numerical increase over the lactation period, the liMac population including liMacs in the tissue and the milk-filled lumen of the alveoli, did not display temporal dynamics in terms of transcription or phenotype, and represented a largely homogenous population.

We found macrophages in both the murine and human milk, and translocation into milk is contrary to the behavior of typical tissue-resident macrophages, which usually do not migrate. The role of macrophages in the milk and their transmission to the infant remain to be investigated. The 3 subsets of milk macrophages in humans had similarities to liMacs, but also to the F4/80[hi] and F4/80[lo] macrophages in the mouse mammary gland. Cells in human breast milk, in particular lactocytes, are highly dynamic during lactation, possibly due to hormonal regulation and health or lifestyle changes. How the three subsets of human milk macrophages change over the course of lactation, and whether there is further diversity in infection remain to be resolved. Total human milk leukocytes increase during infection of the mother or the child[14,47]. It is probable that the number of milk macrophages also correlates with the health status of the mother–infant dyad. We found that liMacs in the murine mammary gland responded rapidly to microbial exposure and, upon LPS challenge played a role in the early recruitment of neutrophils, which are essential for resolving *E. coli*-induced mastitis[38,39]. These observations indicate that liMacs might participate in immune surveillance to protect the mammary gland from invading pathogens and infection.

Macrophages in the mammary gland have been implicated in the development of the mammary gland during pregnancy[7] and its remodeling during involution[4]. We investigated whether liMacs support tissue transformation for lactation. Although we detected transcripts encoding main milk proteins in liMacs, it was unclear whether liMacs expressed milk proteins or whether this was due to phagocytosis of dead epithelial cells, as previously suggested[4,17]. Yet, liMac-deficient mammary glands did not exhibit a changed protein content of the

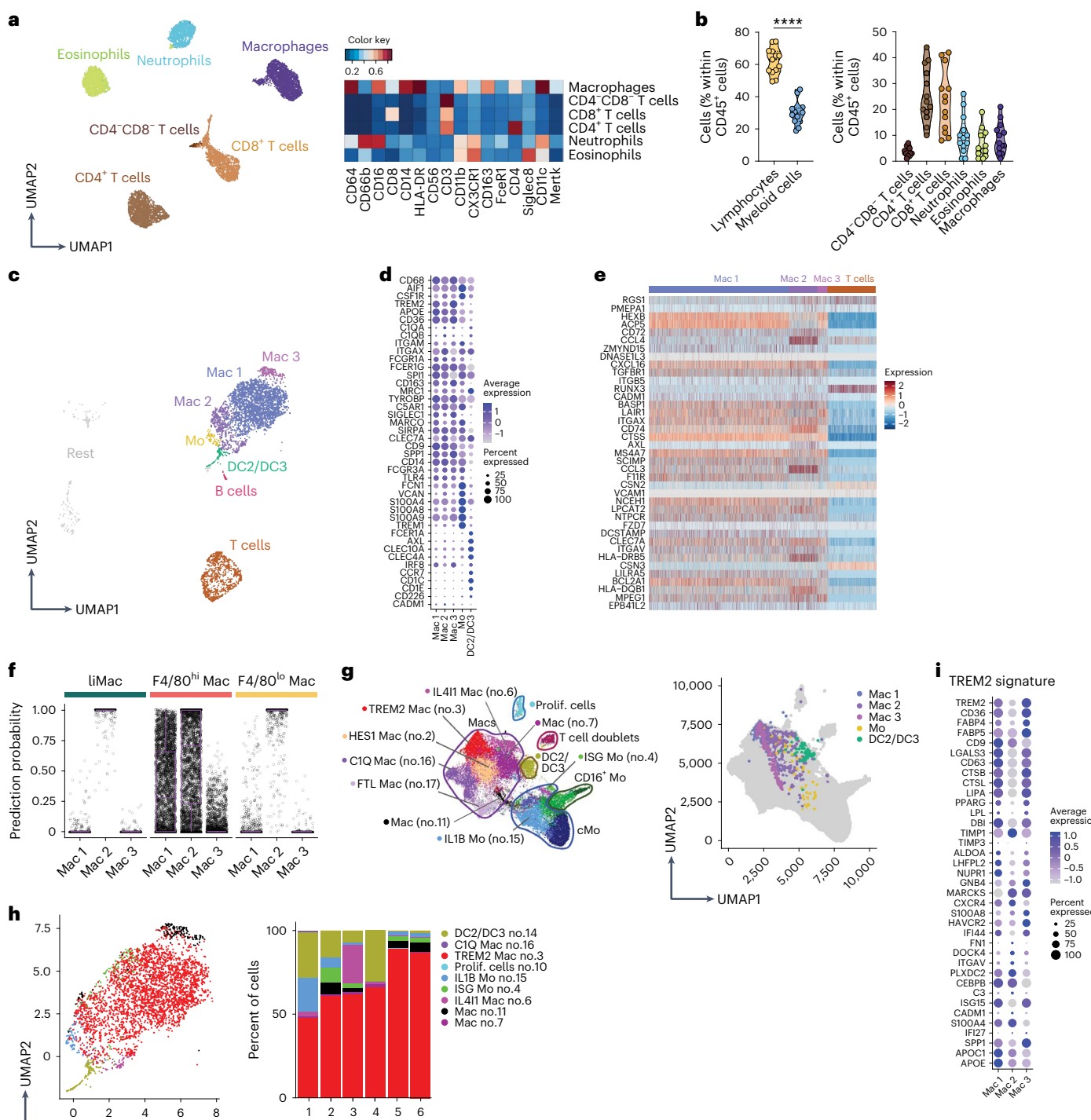

**Fig. 6 | Human milk contains several subsets of macrophages. a,b,** UMAP and corresponding heatmap of mean marker expression (**a**) showing immune cells in human milk (collected on days 4–63 pp and pregated on CD45⁺ cells) and graphs showing percentage of immune cell populations among CD45⁺ cells in human milk (**b**) (n = 13), analyzed by flow cytometry. Representative UMAP is shown from one of four independent experiments, n = 4. Two-tailed unpaired Student's t test was performed. ****P < 0.0001. **c**, Seurat guided clustering and dimensionality reduction by UMAP showing seven immune cell populations including B cells, T cells, three populations of macrophages (Mac 1–3), monocytes (Mo), DC2/DC3 and unidentified clusters termed 'Rest' in scRNA-seq analysis of immune cells (CD45⁺) sorted from human milk samples (n = 6). **d**, Dot plot showing expression of selected genes in the three human macrophage subsets, monocytes and DC2/DC3 as in **c**. Dot size represents percentage of cells in a cluster expressing each

gene; dot color reflects expression level. **e**, Heatmap showing the expression of liMac signature genes (as in Fig. 2c) by human milk macrophages (Mac 1, Mac 2 and Mac 3) and T cells as in **c**. **f**, Prediction similarity scores of mouse mammary gland macrophages applied to human milk macrophage subsets as in **c**. Each dot represents a cell. **g**, UMAP of the MoMac-VERSE compendium with annotated clusters[42] (left) and Mac 1, Mac 2, Mac 3, monocytes (Mo) and DC2/DC3 (as in **c**) projected on the MoMac-VERSE (right). **h**, UMAP and frequencies of the MoMac-VERSE populations projected on human milk Mac 1, Mac 2, Mac 3, monocytes (Mo) and DC2/DC3 (as in **c**). **i**, Dot plot showing expression of selected 'TREM2 signature' genes[22,42,45] in the human macrophage clusters Mac 1, Mac 2, Mac 3 as in **c**. Dot size represents percentage of cells in a cluster expressing each gene; dot color reflects expression level (as indicated on legend).

milk, nor structural alterations in the mammary gland, proposing that liMacs are not involved in regulating tissue integrity during lactation. Macrophages in the milk or the mammary gland may be implicated in lipid metabolism, as suggested by their TREM2 signature, previously identified in lipid-associated macrophages[45]. A similar TREM2 macrophage profile was further associated with immunosuppressive properties in cancer[22,42–44], suggestive of a possible similar role in milk.

Taken together, we describe phenotypically and transcriptionally a murine macrophage population that arises in the mammary gland during lactation and extravasates into the milk, and further identified different subsets of macrophages in human milk. These data will open new avenues for investigating the functions of macrophages for mother and infant during the lactation period, in health and disease.

## Online content

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

## Methods

### Mice

C57BL/6JRj (CD45.2) mice were purchased from Janvier Laboratories. $Csf1r^{fl/fl}$ mice were kindly provided by J. Pollard[48]. $Ms4a3^{Cre}$ mice were kindly provided by F. Ginhoux[26]. $Ccr2^{CreER-mKate}$ ($Ccr2^{CreER}$) mice were generated by Taconic Artemis[27]. $Itgax^{Cre}$ (ref. [49]), $Il34^{LacZ/LacZ}$ (ref. [33]), $Ccr2^{-/-}$ (ref. [50]), $Cx3cr1^{GFP}$ (ref. [51]) and $R26^{Ai14}$ (ref. [52]) were bred inhouse. All 'Cre' and 'CreER' strains were used as heterozygotes. All mice were kept in individually ventilated cages under specific-pathogen-free conditions with a 12 h light-dark cycle, under controlled temperature (21–24 °C) and humidity (35–70%).

Germ-free and Cuatro-colonized C57BL/6J mice were bred and maintained in flexible-film isolators at the Clean Mouse Facility, University of Bern, Switzerland. Hygiene status was monitored routinely by culture-dependent and -independent methods, including 16S rRNA sequencing for Cuatro mice. All mice were confirmed to be pathogen free.

Female mice were used and were normally between 7 and 12 weeks old. Mammary glands of lactating female mice were analyzed between days 1 and 21 pp as indicated in the figure legends; virgin control females were age-matched where possible. No statistical methods were used to predetermine sample sizes but our sample sizes are similar to those reported in previous publications[28]. Data distribution was assumed to be normal but this was not tested formally. No special method of randomization was used. No animals were excluded from analysis. All experimental procedures at the University of Zurich were performed in accordance with Swiss Federal regulations and approved by the Cantonal Veterinary Office of Zurich.

### Cell suspension preparation

Mice were sacrificed by $CO_2$ inhalation and perfused intracardially with PBS (Gibco). After perfusion, left and right abdominal mammary glands were dissected, and inguinal lymph nodes removed. All the samples were cut into small pieces in an Eppendorf tube, followed by digestion in 0.4 mg ml$^{-1}$ collagenase type IV (Worthington) and 0.04 mg ml$^{-1}$ DNase I in $Ca^{2+}/Mg^{2+}$ HBSS supplemented with 10% FCS for 40 mins at 37 °C, while shaking. Samples were homogenized with an 18G needle and a syringe and filtered through a 100 µm cell strainer to obtain a homogeneous cell suspension. Cells were washed once in PBS, resuspended in red blood cell lysis buffer (0.16 M $NH_4Cl$, 0.11 M $KHCO_3$, 0.001 M EDTA in $ddH_2O$) and incubated on ice for 5 min, then filtered through a 100-µm cell strainer and washed with PBS.

### Flow cytometry

Cells were incubated with anti-mouse CD16/32 (clone 93) in PBS for 15 min to block Fc receptors before labeling for 20 min at 4 °C with the antibody mix detailed below diluted in PBS. Data were acquired on BD LSRII Fortessa, BD FACSymphony or Cytek Aurora and analyzed with FlowJo software (Tree Star) and R studio. Cell sorting was performed on BD FACS Aria III and BD S6 equipped with a 100 µm nozzle.

Fluorochrome-conjugated monoclonal antibodies (mAbs) specific for mouse I-A/I-E (clone M5/114.15.2 1:400), CD11b (clone M1/70 1:200), CD11c (clone N418 1:200), CD45 (clone 30-F11 1:400), Ly6C (clones HK1.4 and AL-21 1:400), Ly6G (clone 1A8 1:200), Siglec-F (clone E50-2440 1:200), CD3 (clone 17A2 1:400), NK1.1 (clone PK136 1:200), CD19 (clone 1D3 1:400), CD64 (clone X54-5/7.1 1:100), F4/80 (clones Cl:A3-1 and BM8 1:400), CD169 (clone SER-4 1:200), MerTK (clone DS5MMER 1:200), CD206 (clone C068C2 1:200), Lyve1 (clone ALY7 1:100), CD38 (clone 90 1:400), CX3CR1 (clone SA011F11 1:200), Dectin-1 (clones RH1 and 2A11 1:400) and XCR1 (clone ZET 1:200) were purchased either from BD, eBioscience, Bio-Rad, R&D or Biolegend. Before detailed analysis, cells were gated on single and live cells. Dead cells were excluded with the Fixable Viability Kit (Near-IR staining, Biolegend). mAbs specific for human CD11b (clone ICRF44 1:400), CD11c (clone BU15 1:100),

CD14 (clone M5E2 1:100), CD16 (clone 3G8 1:100), CD64 (clone 10.1 1:50), CD66b (clone G10F5 1:50), CD3 (clone UCHT1 1:100), CD4 (clone RPAT4 1:150), CD8 (clone 3B5 1:100), Siglec8 (clone 7C9 1:50), MerTK (clone 590H11G1E3 1:50), FceR1 (clone CRA1 1:50), CD163 (clone GHI/61 1:50), CX3CR1 (clone 2A9 1:1:50), HLA-DR (clone G46-6 1:200) and CD56 (clone NCAM16.2 1:100) were purchased either from BD or Biolegend. Click-iT EdU Cell Proliferation Kit (ThermoFisher) was used for the cell proliferation assay.

Immunoglobulin concentration in mouse milk was measured by LEGENDplex Mouse Immunoglobulin Isotyping Panel (6-plex, Biolegend) and analyzed with FlowJo software (v.10.6) (BD Bioscience) and LEGENDplex software (v.8.0) (Biolegend).

For flow cytometry high-dimensional analysis, raw data were preprocessed using FlowJo, followed by transformation, percentile normalization, dimensionality reduction and visualization by uniform manifold approximation and projection (UMAP)[53] in R studio. The FlowSOM algorithm was used for automated clustering[54] using UMAP with overlaid marker expression values and a heatmap of median or mean expression values[55–57].

### Milk collection and isolation of leukocytes

Dams were anesthetized with 6.5 mg kg$^{-1}$ bodyweight xylazine (Xylasol, Graeub) and 65 mg kg$^{-1}$ bodyweight ketamine (Ketasol-100, Graeub) injected i.p. Subsequently, dams were stimulated with 4 IU of oxytocin (Sigma–Aldrich) i.p., and milk was expressed manually from the mammary glands and collected using an insulin syringe.

Human milk samples were collected at the University Hospital Zurich from healthy women, aged 20–40 years (mean (S.D.) 32.0 (5.0)), within 4–63 days after delivery. Milk expression from both breasts was performed by donors either at the hospital or at home using an electric pump (Symphony, Medela), a maximum of 12 h before transfer to the laboratory. The collected human milk was homogenized in the hospital, 1–3 ml were held for the present analysis while the rest was used for infant feeding. The study protocol was approved by the Swiss Ethics Commission of the Canton of Zurich (BASEC-Nr. 2020-00542) and all the subjects participating in the study signed informed consent before enrollment.

For isolation of milk cells, milk was diluted in a 1:1 ratio with PBS and centrifuged at 800$g$ for 20 min at 15 °C. The lipid layer and skim milk were removed, and the cell pellet was washed twice in PBS by centrifugation at 400$g$ for 5 min and resuspension in PBS.

### Antibody treatment

CSF-1 neutralizing antibody (clone 5A1) and isotype control antibody (rat IgG1, clone HPRN) were purchased from Bio X Cell, and were administered i.p. at a dose of 0.2–0.3 mg.

### Tamoxifen treatment

Tamoxifen was reconstituted with 100% ethanol in corn oil with a final concentration of 25 mg ml$^{-1}$. A total of 5 mg of tamoxifen was administered via oral gavage to pregnant mice.

### LPS-induced mastitis

CSF-1 neutralizing antibodies (clone 5A1) were administered i.p. 1 and 2 days before LPS injection. Mice were anesthetized with isoflurane and the fourth mammary glands were injected with 5–10 µg of LPS (Sigma); 18 h later the mice were sacrificed and mammary glands were analyzed.

### Phagocytosis assay

Mammary gland cells were isolated as described above. Following isolation, cells were stained as described above, resuspended in RPMI with 2% FBS and incubated with pHrodo Red $E. coli$, Zymosan and $S. aureus$ BioParticles (Invitrogen) (0.01 mg ml$^{-1}$, 0.05 mg ml$^{-1}$ and 0.01 mg ml$^{-1}$, respectively), for 1.5 h at 37 °C and 5% $CO_2$. After the incubation, cells were immediately acquired.

## In vitro stimulation with TLR ligands

Cells were isolated as described above from inguinal, abdominal and thoracic mammary glands. Following isolation, cells were resuspended in full medium (RPMI (Seraglob), 10% FBS (Gibco), 0.01 M HEPES (Gibco), 1 mM sodium pyruvate (Gibco), 2 mM Glutamax (Gibco), 1% Penicillin–Streptomycin (Gibco), 1% nonessential amino acids (Gibco), 57.2 μM β-mercaptoethanol (AppliChem)) with GolgiPlug and GolgiStop 1:1,000 (BD) and exposed to Zymosan 50 μg ml⁻¹ (InvivoGen) or LPS 300 ng ml⁻¹ (Sigma) or left unstimulated (negative control). The cells were incubated for 6 h at 37°C, washed and stained for surface markers as described above, then fixed with Cytofix/Cytoperm (BD Biosciences) for 20 min at 4°C, washed with Perm buffer (2% BSA, 0.5% Saponin, 0.0002% sodium azide in PBS), stained with intracellular antibody mix in Perm buffer for 25 min (TNF (clone MP6-XT22 1:400) and proIL-1β (clone NJTEN3 1:200), purchased either from eBioscience or Biolegend), washed with Perm buffer, resuspended in PBS and acquired.

## Histology

Mice were euthanized through $CO_2$ asphyxiation and perfused with PBS. Mammary fat pads were removed, fixed in 4% PFA (Morphisto) for 6–24 h at 4 °C, washed in PBS followed by incubation in 30% sucrose in PBS at 4 °C for 1–5 days. The tissue was then embedded in Cryo Embedding Medium (Medite) and frozen on dry ice.

Sections (20 μm) were cut using a Hyrax C60 cryostat (Zeiss) onto glass slides. Sections were dried for 20 min at 37 °C, washed twice with PBS, and blocked for 2 h at room temperature in 10% normal goat serum (ThermoFisher Scientific) and 0.5% Triton X-100 (Sigma–Aldrich) in PBS. For some stainings, sections were then blocked additionally for 45 min at room temperature with the M.O.M kit Mouse Ig Blocking Reagent (90 μl in 2.5 ml PBS; Vector Laboratories). Primary antibody was applied overnight at 4 °C in 5% normal goat serum and 0.5% Triton X-100 in PBS, using the following antibodies: mouse anti-smooth muscle actin (Dako, 1:25), Armenian hamster anti-CD11c conjugated to AlexaFluor 594 (BioLegend, 1:200), rat anti-Lyve1 conjugated to eFluor 660 (eBioscience, 1:100) and rat anti-Ki67 conjugated to PE-Cy7 (eBioscience, 1:500). Secondary antibody (goat anti-mouse IgG conjugated to AlexaFluor 405, ThermoFisher Scientific, 1:200) was then applied for 2 h at room temperature in 5% normal goat serum and 0.5% Triton X-100 in PBS. After labeling, sections were mounted in mounting medium, with or without DAPI (Dianova), under glass coverslips. Images were acquired on an SP8 confocal microscope (Leica), using a ×20 multi-immersion objective or a ×63 oil objective. LAS X (Leica) and Imaris (Bitplane) software was used for analysis.

Sections 60 μM (free-floating) were cut using a Hyrax C60 cryostat (Zeiss) and subsequently permeabilized by incubation in blocking buffer (PBS + 2% normal goat serum + 0.1% Triton X-100) for 1 h at RT. Free-floating sections were then stained with primary antibodies at 4 °C for 2–3 days using anti-Iba1 (1:500, Wako), anti-GFP (1:200 Nacalai Tesque), anti-SMA (1:100, Sigma), anti-CD64 (1:100, Bio-Rad), anti-Dectin-1 (1:100, Bio-Rad), anti-CD138 (1:100, BioLegend), anti-IgA (1:100, Southern Biotech) or anti-MHCII (1:100, BioLegend). After washing three times with PBS for 5 min each, sections were incubated for 2 h at RT with DAPI (1:1,000 ThermoFisher) and secondary antibodies diluted 1:500. Afterwards, the washing steps were repeated and sections were mounted with some drops of mounting medium (Dianova). High-resolution images were acquired on a SP5 upright or SP8 Falcon confocal microscope (Leica) with ×20 and ×40 objectives. Images were processed using the Imaris software (Bitplane).

For H&E staining, either frozen sections were taken or paraffin-embedded sections previously deparaffinized in xylene and rehydrated in an ethanol gradient of 100%, 95% and 70%. Subsequently, sections were rinsed in water, stained with hematoxylin (Sigma) for 5 mins, rinsed with water, counterstained with eosin (Morphisto) for 1 min, dehydrated in 95%, 100% ethanol gradient, cleared with xylene

and mounted with Eukitt mounting medium (Sigma). Sections were then imaged on a brightfield microscope (Olympus), and images were segmented with the ilastik software[58] using a pixel classification workflow and subsequently exported to Fiji (National Institutes of Health (NIH)). The number and size of alveoli were calculated using the Analyze Particles command. Only alveoli with areas between 1,000 pixels (458 μm²) and 50,000 pixels (22,888 μm²) were included in the summary statistics, to exclude small, nonspecific speckles, and large aggregations of alveoli that had not been properly segmented one from another.

## Single-cell RNA sequencing and data analysis

Murine cells were barcoded with TotalSeq anti-mouse Hashtag antibodies B0304, B0305 and B0306 (clones M1/42; 30-F11) (for days 8, 11 and 14 pp). Human milk cells were barcoded with TotalSeq anti-human Hashtag antibodies B0251, B0252, B0253, B0254 and B0255 (clones LNH-94 and 2M2). For murine cells, 10,000 (for day 7 pp, $n = 1$ for virgin mammary gland and $n = 2$ for lactating mammary gland) and 18,000 cells (for days 8, 11, 14 pp, $n = 3$ for each timepoint), and for human cells, 10,000 cells (8–63 days pp, $n = 6$) were loaded into 10x Genomics Chromium, and library preparation was performed according to the manufacturer's instructions (Single Cell 3' v.3 protocol).

The libraries were sequenced on a NovaSeq6000 platform with a depth of around 50,000 reads per cell. CellRanger (v.3.1.0) from 10x Genomics was used to demultiplex, align the reads to GENCODE reference build GRCm38.p6 Release M23 (murine cells) and collapse unique molecular identifiers. The human milk sample was aligned to GRCh38.p13.

Starting from the filtered gene-cell count matrix produced by CellRanger's inbuilt cell-calling algorithms, samples were analyzed using the Seurat v.3 workflow. In brief, low quality cells were filtered according to genes detected, unique molecular identifiers counted and mitochondrial content. As an additional filtering step, doublets were removed using the scDblFinder package[59]. Cell cycle state was determined using the cyclone algorithm[60]. The three samples at timepoint p7 were integrated by the fastMNN method with 2,000 variable features. The cells were visualized using UMAP dimension reduction. Cluster identity was determined manually based on the marker genes computed by Seurat's FindMarkers function. The selected primary clusters were reclustered individually. After subcluster identity identification, subcluster labels were merged for the final cell annotation. Gene ontology annotations were downloaded from the Ensembl database, and GO enrichment of gene sets was computed using the clusterprofiler package. Differentially expressed genes between days 8, 11 and 14 pp were calculated using the limma voom method, implemented in the volcano3D package via the voom polar function. Volcano plots were generated with volcano3D function with cut off at 0.01 and filter pairwise set to false. For the multimodal MoMac-VERSE reference mapping, the multimodal reference mapping pipeline from Seurat was used[61] with same number of PC (50) as in the original paper[42].

For comparison of the human milk sample with mouse data, human genes were mapped to the orthologous mouse genes, and subsequently integrated using Seurat's Integrated Data functionality (PCA). To compare murine and human macrophage populations with similarity score, the Seurat V3 Integration and Label Transfer pipeline with the human data as reference was used. Count matrixes and metadata from both datasets were SCT transformed, and integration anchors were extracted with default parameters[62]. Human macrophage annotations were transferred to murine cells using the TransferData function with 30 k-nearest neighbor dimensions (Similarity Score)[22,63].

## Proteomics analysis

Sample preparation: 80 μl per sample was mixed with 100 μl of SDS buffer (4% SDS, 100 mM Tris-HCl pH 8.2, 0.1 M dithiothreitol), boiled at 95 °C for 5 min and processed with high intensity focused ultrasound

for 90 s. Proteins were then diluted in 800 µl of UT buffer (urea 8 M in 100 mM Tris-HCl pH 8.2), loaded on an Ultracel 30,000 molecular weight cut off centrifugal unit (Amicon Ultra, Merck) and centrifuged at 14,000 g. SDS buffer was exchanged by one centrifugation round of 200 µl UT buffer. Alkylation of reduced proteins was carried out by 5 min incubation with 100 µl iodoacetamide 0.05 M in UT buffer, followed by three 100 µl washing steps with UT and three 100 µl washing steps with NaCl 0.5 M. Finally, proteins were digested on filter using 120 µl of 0.05 triethylammonium bicarbonate buffer (pH 8) containing trypsin (Promega) at a ratio of 1:50 (w/w), adapted from the filter-aided sample preparation (FASP) protocol[64]. Digestion was performed overnight in a wet chamber at room temperature. After elution, the solution containing peptides was acidified to a final 0.1% trifluoroacetic acid, 3% acetonitrile concentration. Peptides were desalted using inhouse C18 stage tips, dried and resolubilized in 20 µl of 3% acetonitrile, 0.1% formic acid for mass spectrometry (MS) analysis. Peptide concentration was estimated using nanodrop and the volumes normalized to a peptide concentration of 0.6 µg µl⁻¹.

### Liquid chromatography-MS analysis

Mass spectrometry (MS) analysis was performed on an Orbitrap Fusion Lumos (ThermoFisher Scientific) equipped with a Digital PicoView source (New Objective) and coupled to a M-Class UPLC (Waters). Solvent composition at the two channels was 0.1% formic acid for channel A and 0.1% formic acid, 99.9% acetonitrile for channel B. Column temperature was 50 °C. For each sample 2 µl of peptides was loaded on a commercial ACQUITY UPLC M-Class Symmetry C18 Trap Column (100 Å, 5 µm, 180 µm × 20 mm, Waters) followed by ACQUITY UPLC M-Class HSS T3 Column (100 Å, 1.8 µm, 75 µm × 250 mm, Waters). The peptides were eluted at a flow rate of 300 nl min⁻¹. After a 3 min initial hold at 5% B, a gradient from 5% to 22% B in 83 min and 22% to 32% B in an additional 10 min was applied. The column was cleaned after the run by increasing to 95% B and holding 95% B for 10 min before reestablishing the loading condition. Samples were acquired in random order. The mass spectrometer was operated in data-dependent mode acquiring a full-scan MS spectrum (300−1,500 m/z) at a resolution of 120,000 at 200 m/z after accumulation to a target value of 500,000. Data-dependent MS/MS were recorded in the linear ion trap using quadrupole isolation with a window of 0.8 Da and higher-energy C-trap dissociation fragmentation with 35% fragmentation energy. The ion trap was operated in rapid scan mode with a target value of 10,000 and a maximum injection time of 50 ms. Only precursors with intensity above 5,000 were selected for MS/MS and the maximum cycle time was set to 3 s. Charge state screening was enabled. Singly, unassigned, and charge states higher than seven were rejected. Precursor masses previously selected for MS/MS measurement were excluded from further selection for 20 s, and the exclusion window was set at 10 ppm. The samples were acquired using internal lock mass calibration on m/z 371.1012 and 445.1200. The MS proteomics data were handled using the local laboratory information management system[65]. Data are available at ProteomeXchange with identifier PXD041711 (ref. 66).

The acquired raw MS data were processed by MaxQuant (v.1.6.2.3), followed by protein identification using the integrated Andromeda search engine. Spectra were searched against a Uniprot mouse reference proteome (taxonomy ID 10090, proteome ID UP000000589), concatenated to its reversed decoyed fasta database and common protein contaminants. Carbamidomethylation of cysteine was set as fixed, while methionine oxidation and N-terminal protein acetylation were set as variable modifications. Enzyme specificity was set to trypsin/P, allowing a minimal peptide length of seven amino acids and a maximum of two missed cleavages. MaxQuant Orbitrap default search settings were used. The maximum false discovery rate was set to 0.01 for peptides and 0.05 for proteins. Label-free quantification was enabled, and a 2-min window for match between runs was applied. In the

MaxQuant experimental design template, each file was kept separate in the experimental design to obtain individual quantitative values. We used protein intensity values that were generated in the MaxQuant and were reported in proteinGroups.txt file. Preprocessing of the protein intensities was performed as follows: intensities equal to zero were removed, while nonzero intensities were log₂ transformed and modified using robust z-score transformation to remove systematic differences between samples. To estimate log₂ fold changes and P values, the Bioconductor package limma was used. P values for multiple testing were adjusted using the Benjamini and Hochberg procedure and false discovery rates were obtained.

### Statistical analysis

Statistical analysis, including P values and mean ± s.d. or ± s.e.m. (where applicable) was performed using GraphPad Prism v.7 and v.9 (GraphPad Software); n represents number of biological replicates. Statistical details for each experiment can be found in the corresponding figure legends.

### Reporting summary

Further information on research design is available in the Nature Portfolio Reporting Summary linked to this article.

## Data availability

The mouse mammary gland and human milk scRNA-seq datasets were deposited at the Genome Expression Omnibus under the accession numbers GSE230697 and GSE230749, respectively. The MS proteomics data were deposited onto the ProteomeXchange Consortium via the PRIDE partner repository with the dataset identifier PXD041711.

## Code availability

This study did not generate new codes. The codes that support these findings have been previously described[42,56,57]. The analysis code for the MoMac-VERSE analysis was uploaded to Github (https://github.com/gustaveroussy/FG-Lab).

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

## Acknowledgements

We thank the Flow Cytometry Facility (University of Zurich) and the Center for Microscopy and Image Analysis (University of Zurich) for technical support; E. Yángüez, D. Popovic and D. Ehrsam for performing single-cell RNA sequencing, D. González Rodríguez for bioinformatics support and P. Nanni for proteomics analysis at the Functional Genomics Center Zurich (FGCZ) and M. Lutz (Institute of Experimental Immunology, University of Zurich) for technical support. We thank the staff of the Clean Mouse Facility of the University of Bern for technical support. We are grateful for the support of the Family Larsson-Rosenquist Foundation. We also thank L. Robinson of Insight Editing London for assistance reviewing the manuscript. This work was supported by grants from the Swiss National Science Foundation (310030_184915, PP00P3_170626 and BSSGI0_155832 to M.G., 310030_146130, 316030_150768, 310030_170320 to B.B. and CRSII5_177164 to A.J.M.). This project has received funding from the European Research Council (ERC) under the European Union's Horizon 2020 research and innovation program (grant agreement No. 819229 to M.G.).

## Author contributions

D.C. and E.P. performed experiments, analyzed data, designed the study and wrote the manuscript. G.T., H.R. and V.K. performed bioinformatics analysis. W.M. and C.A.W. performed histological analysis. S.K., C.M., E.R., S.A.S., M.A., P.Z. and N.P.J. performed experiments. F.G., Z.L., C.B. and K.M shared fate-mapping strains and performed data analysis of human macrophages. F.R. and A.J.M. performed experiments with germ-free mice. M.G. wrote and edited the manuscript, conceptualized, supervised and funded the study. G.N. was in charge of participant data collection. B.B., F.G., G.N. and A.J.M. edited and reviewed the manuscript. S.K. and C.M. contributed equally. All authors read and approved the manuscript.

## Competing interests

The authors declare no competing interests.

## Additional information

**Extended data** is available for this paper at https://doi.org/10.1038/s41590-023-01530-0.

**Correspondence and requests for materials** should be addressed to Melanie Greter.

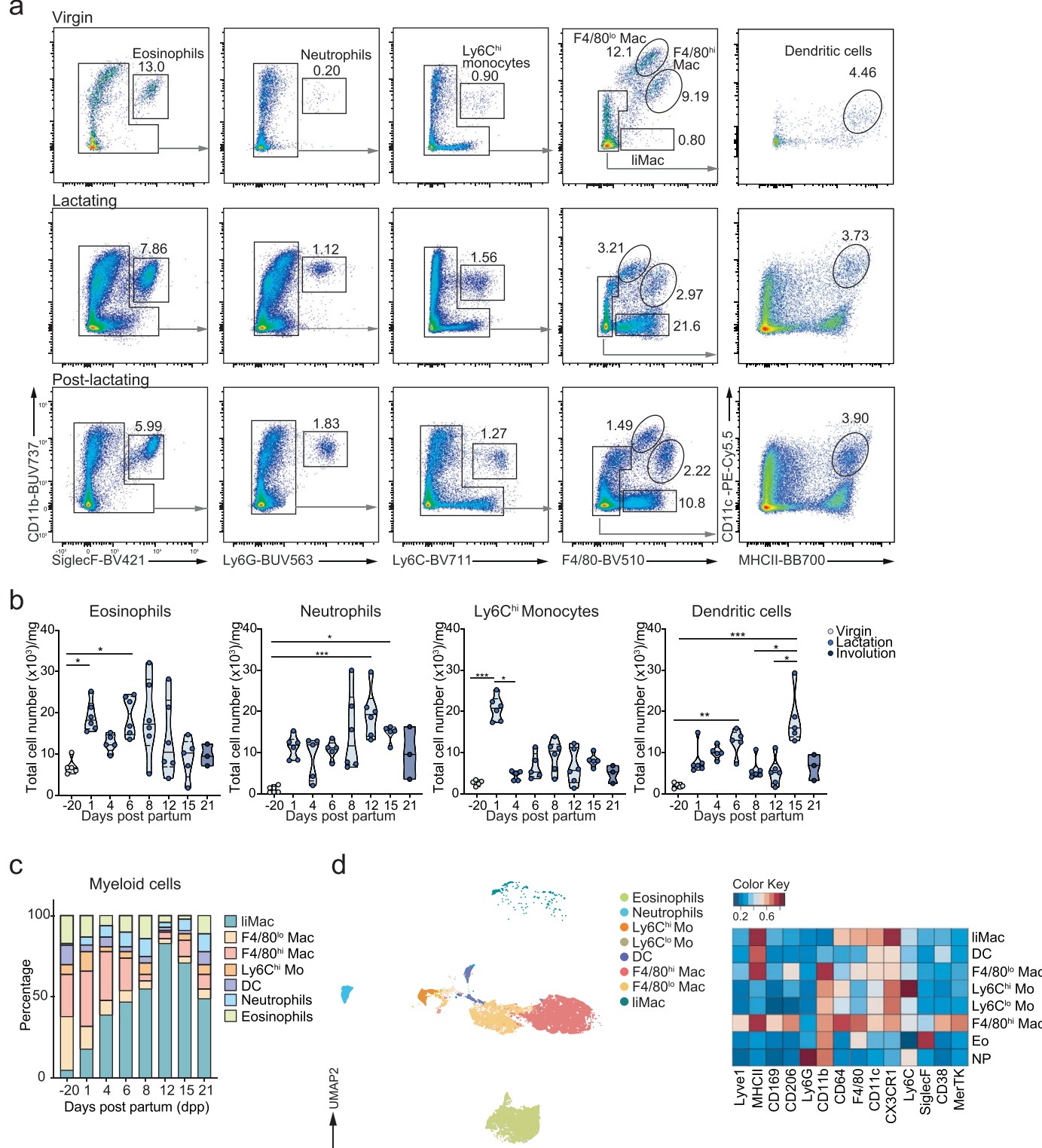

**Extended Data Fig. 1 | Gating strategy and population dynamics of myeloid cells in the murine mammary gland during lactation. a-c**, Representative flow cytometry plots for the gating strategy of eosinophils, neutrophils, Ly6C[hi] monocytes, F4/80[lo] macrophages (Mac), F4/80[hi] macrophages, liMacs and dendritic cells (DC) (pre-gated on live, singlets and CD45[+] cells) (a), violin plots of total cell counts (b) and bar graph with the percentages of total myeloid cells (c) in mammary glands from virgin, lactating (day 10 pp) and post-lactating (day 21 pp) wild-type mice. Data (n = 5-6 per timepoint) are pooled from 6 independent experiments. Kruskal-Wallis test was corrected with Dunn's multiple comparisons test, *p < 0.05; **p < 0.01; ***p < 0.001. **d**, UMAP plot (pre-gated on CD45[+]CD11b[+] and/or CD11c[+]) showing the myeloid compartment of the mammary glands in late gestation (at E18.5), analyzed by flow cytometry. Heatmap shows the mean marker expression. Data were transformed and percentile normalized. Related to Fig. 1.

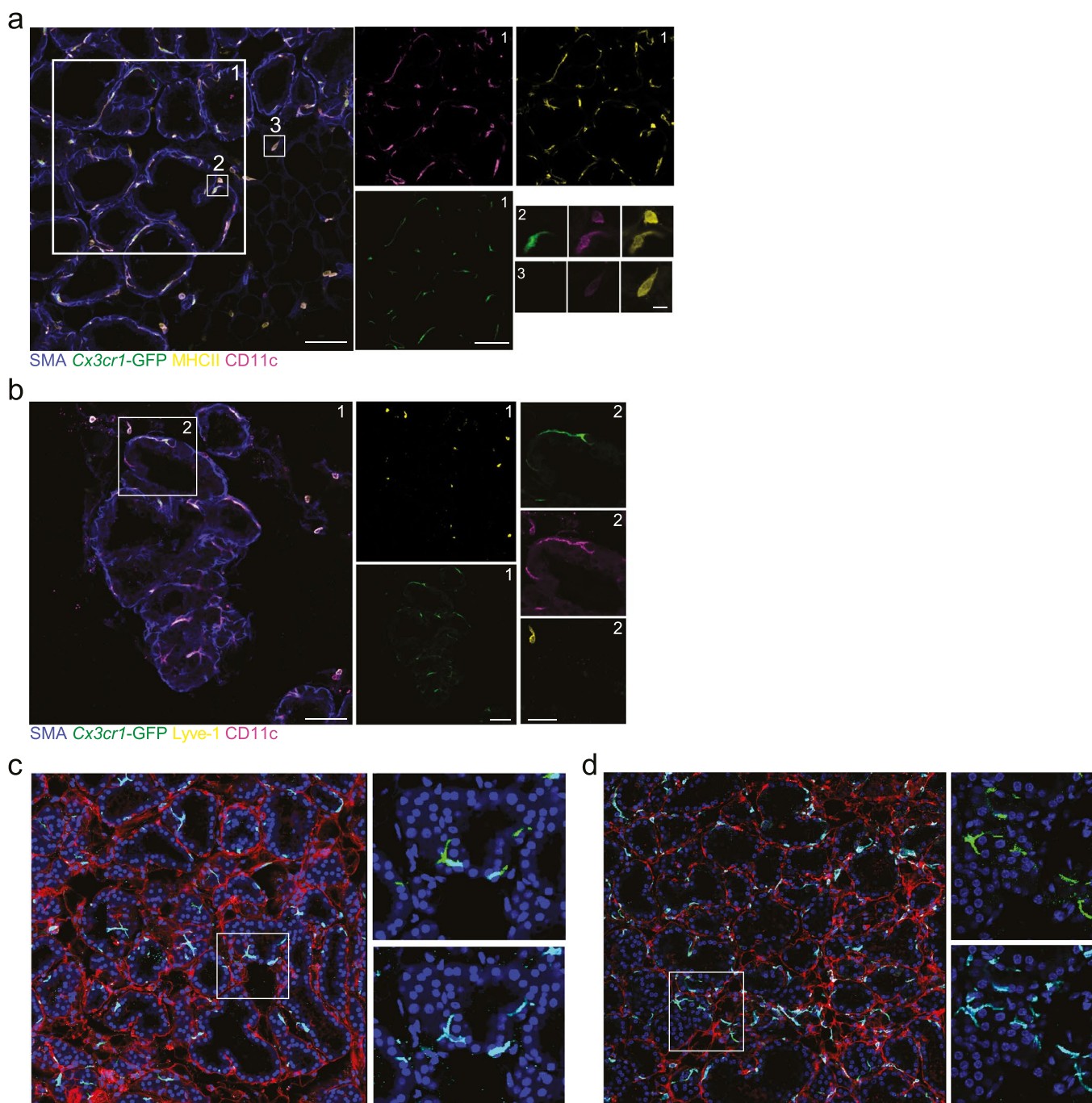

**Extended Data Fig. 2 | LiMacs are localized at the site of milk production in mice. a-b,** Immunofluorescence images of lactating mammary glands from *Cx3cr1*GFP/+ dams (day 8 pp) showing SMA (blue), CD11c (magenta), *Cx3cr1*-GFP (green) and MHCII (yellow) (**a**) or Lyve1 (**b**) (yellow). Insets are magnifications of outlined regions showing single stainings for *Cx3cr1*-GFP, CD11c, MHCII (a) or *Cx3cr1*-GFP, CD11c, Lyve1 (b). Images are representative of n = 2 mice. Scale bar: 75 μm for low magnification, 10 μm (a) or 38 μm (b) for insets.

**c-d,** Immunofluorescence images of lactating mammary glands from *Cx3cr1*GFP/+ dams (day 9–12 pp) showing DAPI (blue), SMA (red), *Cx3cr1*-GFP (green), and CD64 (cyan) (**c**) or Iba1 (cyan) (**d**). Insets are magnifications of outlined regions showing single stainings for *Cx3cr1*-GFP and CD64 (c) and Iba1 (d). Scale bar: 100 μm for low magnification, 50 μm for insets. Images are representative of n = 4 mice. Related to Fig. 1.

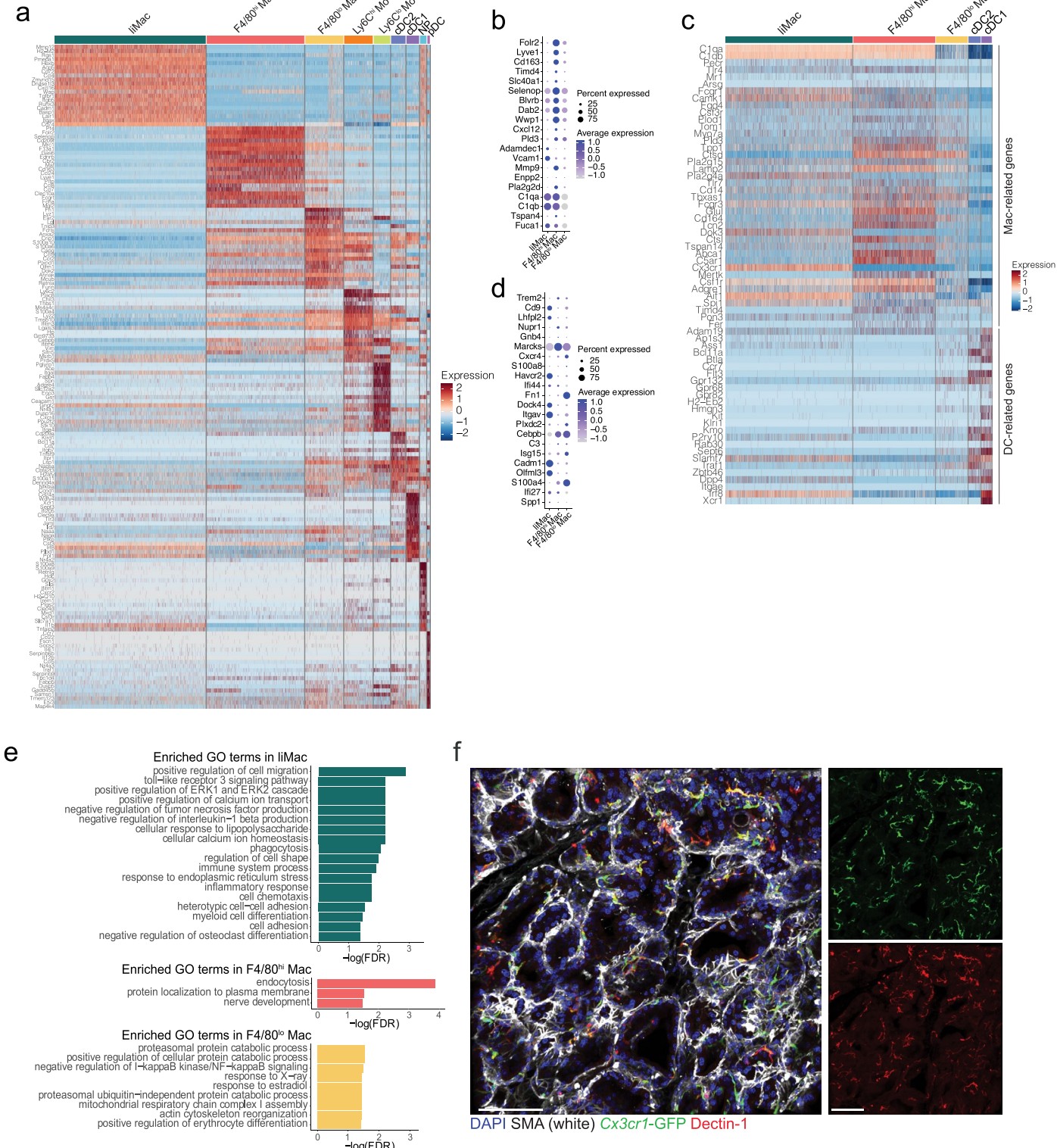

**Extended Data Fig. 3 | Genes highly expressed in liMacs. a-e**, scRNA-seq analysis of myeloid cells (CD11b⁺ and/or CD11c⁺) of lactating mammary glands derived from WT mice at 7 dpp. **a**, Heatmap of cluster-defining genes. Related to Fig. 2a. **b**, Dot plot showing expression of selected 'Folr2 signature' genes[22] in liMacs, F4/80^hi macrophages (Mac) and F4/80^lo macrophages. Dot size represents percentage of cells in a cluster expressing each gene; dot color reflects expression level (as indicated in the legend). **c**, Heatmap showing expression of macrophage and DC signature genes[67,68] in liMacs, F4/80^lo macrophages, F4/80^hi macrophages, cDC2s and cDC1s. **d**, Dot plot showing expression of selected

'Cadm1/Trem2 signature' genes[22] in liMacs, F4/80^hi macrophages and F4/80^lo macrophages. Dot size represents percentage of cells in a cluster expressing each gene; dot color reflects expression level (as indicated in the legend). **e**, Enrichment of GO terms according to the gene expression pattern in liMacs, F4/80^hi macrophages and F4/80^lo macrophages of lactating mammary glands. **f**, Immunofluorescence image of lactating mammary glands from *Cx3cr1*^GFP/+ dams (day 8 pp) showing DAPI (blue), SMA (white), *Cx3cr1*-GFP (green) and Dectin-1 (red) (left) and single stainings for *Cx3cr1*-GFP and Dectin-1 (right). Images are representative of n = 2 mice. Scale bar: 100 μm.

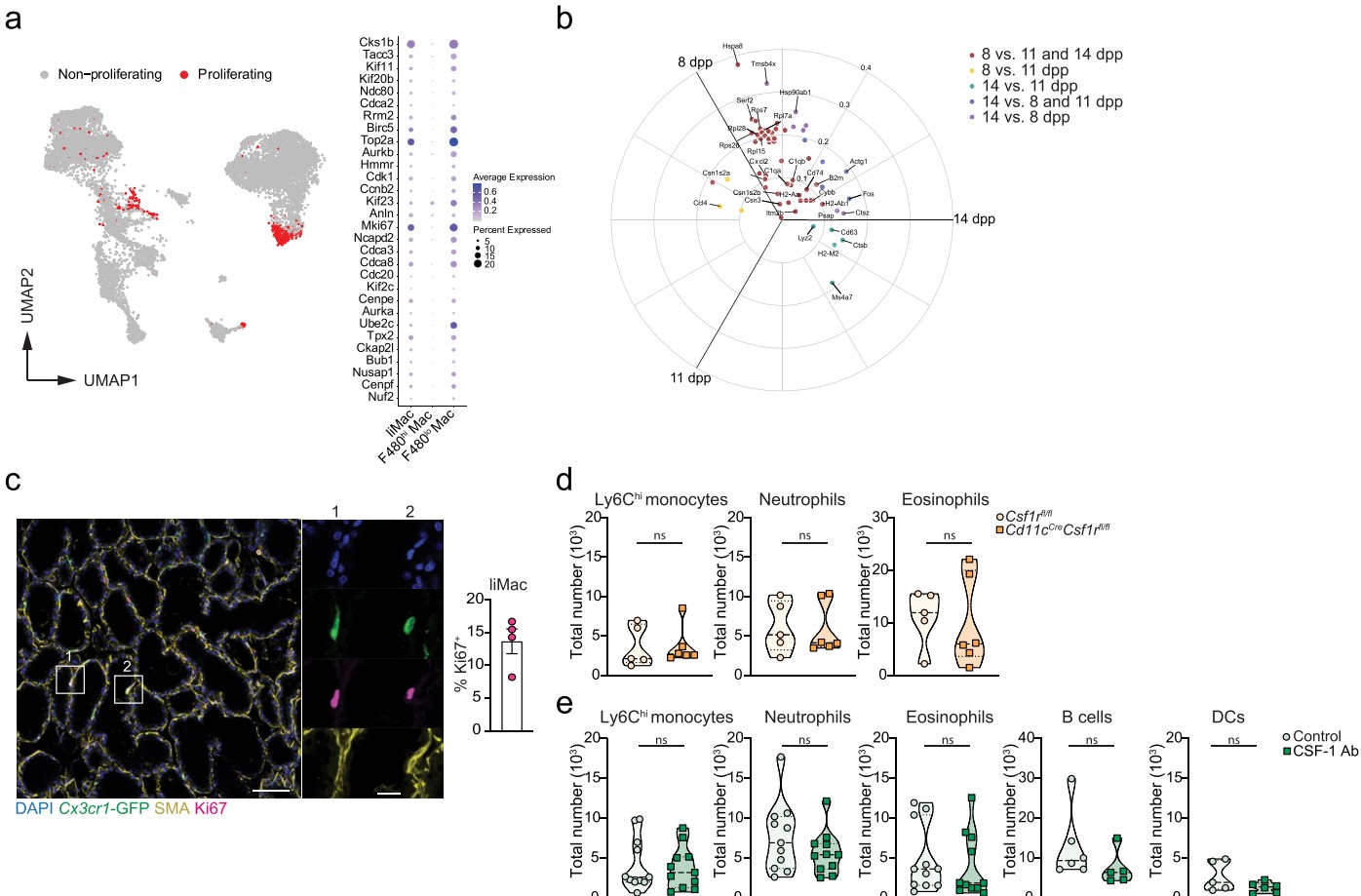

**Extended Data Fig. 4 | LiMacs proliferate during the lactation period in mice. a**, UMAP showing proliferating cells (in red) of scRNA-seq data of myeloid cells (CD11b⁺ and/or CD11c⁺) and dot plot showing expression of proliferation-associated genes in lactating mammary glands (at day 7 pp) of wild-type mice. Dot size indicates percentage of cells expressing the gene; color indicates the average gene expression level per cell. Related to Fig. 2. **b**, Polar volcano plot showing differentially expressed genes between liMacs at days 8, 11 and 14 pp, analyzed by the limma voom method. Coordinates correspond to relative mean expression values per group. 86 genes out of 8052 resulted differentially expressed, of which 46 were specifically upregulated in day 8 pp (adjusted p values < 0.01). 30 genes with top p values are labeled. Related to Fig. 2. **c**, Representative immunofluorescence image and a bar graph (±SEM) showing percentage of Ki67⁺ liMacs in lactating mammary glands from *Cx3cr1*GFP/+ dams

(day 8–10 pp). Ki67 (magenta), SMA (yellow), *Cx3cr1*-GFP (green) and DAPI (blue). Insets are magnifications showing single stainings of the outlined regions. N = 4 mice. Scale bar: 75 μm, inset 18.75 μm. **d**, Violin plots showing total cell counts of Ly6Chi monocytes, neutrophils and eosinophils in lactating mammary glands (day 8–14 pp) of *Csf1r*fl/fl and *Cd11c*Cre*Csf1r*fl/fl dams. Pooled data from 5-6 mice analyzed in 4 independent experiments. Two-tailed Mann-Whitney test was performed, ns = not significant. Related to Fig. 4. **e**, Violin plots showing total cell counts of Ly6Chi monocytes, neutrophils, eosinophils, B cells and DCs in lactating mammary glands (day 7–14 pp) of control (isotype control antibody-treated or untreated) or CSF-1 antibody (Ab)-treated (2-3 times) from wild-type dams. Pooled data from 11 mice analyzed in 5 independent experiments. Two-tailed Mann-Whitney test was performed, ns = not significant. Related to Fig. 4.

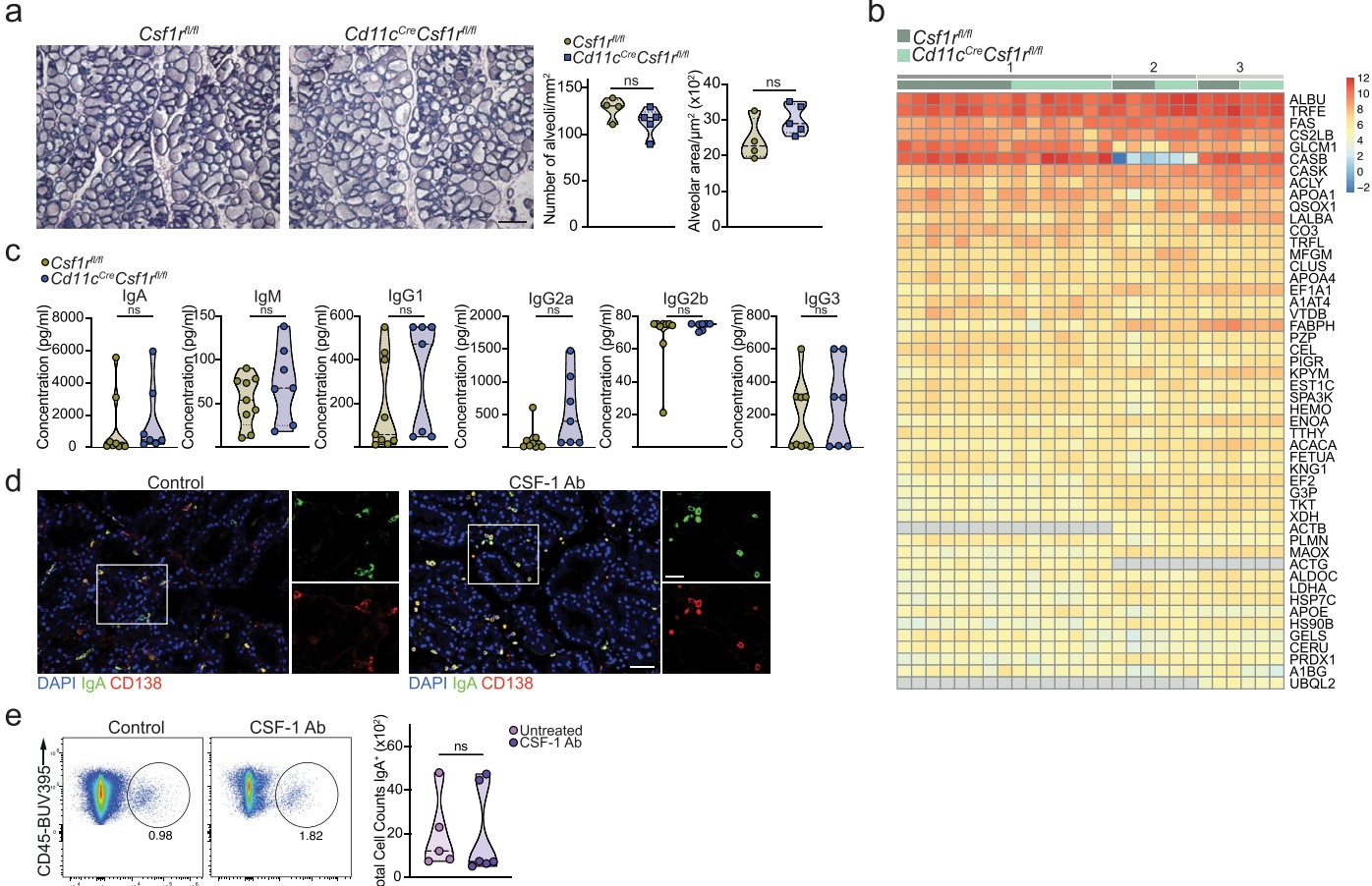

**Extended Data Fig. 5 | Absence of liMacs does not alter protein composition of murine milk. a**, Hematoxylin and eosin staining of mammary gland sections of lactating mammary glands (day 8 pp) from *Csf1r*fl/fl and *Cd11c*Cre*Csf1r*fl/fl dams. The number of alveoli and the alveolar area were quantified using ilastik software. Images are representative of n = 4 and 5 mice, respectively. Scale bar: 200 μm. Two-tailed unpaired Student's t test was performed, ns = not significant. Related to Fig. 5. **b**, Heatmap showing the 50 most abundant proteins in milk derived from *Csf1r*fl/fl and *Cd11c*Cre*Csf1r*fl/fl dams at day 7 pp. N = 13-14, pooled data from 3 individual experiments, indicated with a gray line. **c**, Violin plots showing concentrations of IgA, IgM, IgG1, IgG2a, IgG2b, IgG3 in mouse milk collected at day 7 pp from lactating *Csf1r*fl/fl and *Cd11c*Cre*Csf1r*fl/fl dams. N = 9 and 7 mice,

respectively. Two-tailed unpaired Student's t test was performed, ns = not significant. **d**, Representative immunofluorescence images of lactating mammary glands (day 11–14 pp) of control (untreated) or CSF-1 antibody-treated (on day 8–11 pp) dams showing DAPI (blue), IgA (green) and CD138 (red). Insets are magnifications showing single stainings of the outlined regions. Scale bar: 50 μm in the overview image, 37.5 μm in the inset. N = 2-3 mice. **e**, Representative flow cytometry plots and violin plots showing total cell counts of IgA+ cells in lactating mammary glands (day 7 pp) of isotype-treated or CSF-1 antibody-treated (on day 1, 3, 5 pp) dams. Data were pooled from 3 independent experiments, n = 5-6. Two-tailed Mann-Whitney test was performed, ns = not significant.

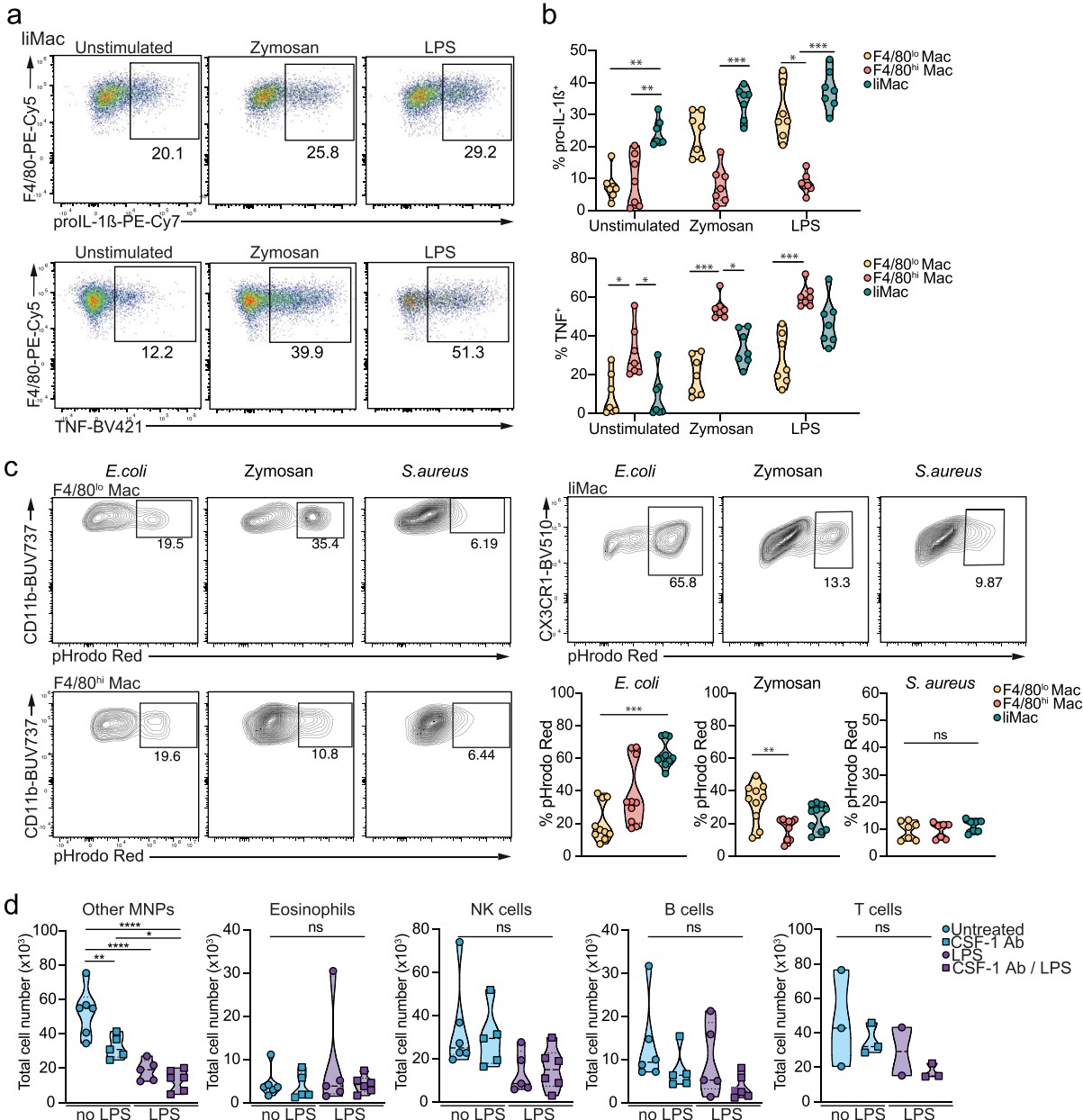

**Extended Data Fig. 6 | The three mammary gland macrophages respond differently to microbial stimuli. a-b**. Representative flow cytometry plots (a) and violin plots (b) showing the percentage of proIL-1β⁺ or TNF⁺ liMacs (a,b) and F4/80ˡᵒ macrophages (Mac) and F4/80ʰⁱ macrophages (b) from lactating mammary glands (day 11–14 pp) of wild-type mice after exposure to LPS or zymosan *in vitro* for 6 h. Data was pooled from two independent experiments, n = 7. Each value is a mean of 3 replicates. Kruskal-Wallis test with Dunn's multiple comparisons test was performed, *p < 0.05; **p < 0.01; ***p < 0.001. **c**, Representative flow cytometry plots and violin plots showing the percentage of pHrodo Red⁺ F4/80ˡᵒ macrophages, F4/80ʰⁱ macrophages and liMacs from the lactating mammary glands (day 10–15 pp) of wild-type mice after exposure *in vitro* to *E. coli*, zymosan and *S. aureus* pHrodo Red bioparticles for 90 minutes. N = 10 for *E. coli* and zymosan pHrodo Red bioparticles, pooled from 3

independent experiments (day 10–15 pp); n = 7 for *S. aureus* pHrodo Red bioparticles, pooled from 2 independent experiments (day 12–15 pp). All samples were run in triplicates and the means were calculated. Kruskal-Wallis test with Dunn's multiple comparisons test was performed, *p < 0.05; **p < 0.01; ***p < 0.001, ns = not significant. **d**, Violin plots showing total cell counts of other mononuclear phagocytes (MNPs) (CD45⁺CD19⁻NK1.1⁻CD3⁻SiglecF⁻Ly6G⁻'LiMac⁻'), eosinophils, NK cells, B cells and T cells in the lactating mammary glands (day 10–14 pp) of control (untreated) and CSF-1 antibody-treated (on day 7–11 pp) mice, challenged with LPS 18 hours prior to analysis or left untreated. Data (n = 5-6 per group) were pooled from 2 independent experiments. One-way ANOVA test was applied, *p<0.05, **p<0.01, ****p<0.0001, ns = not significant. Related to Fig. 5.

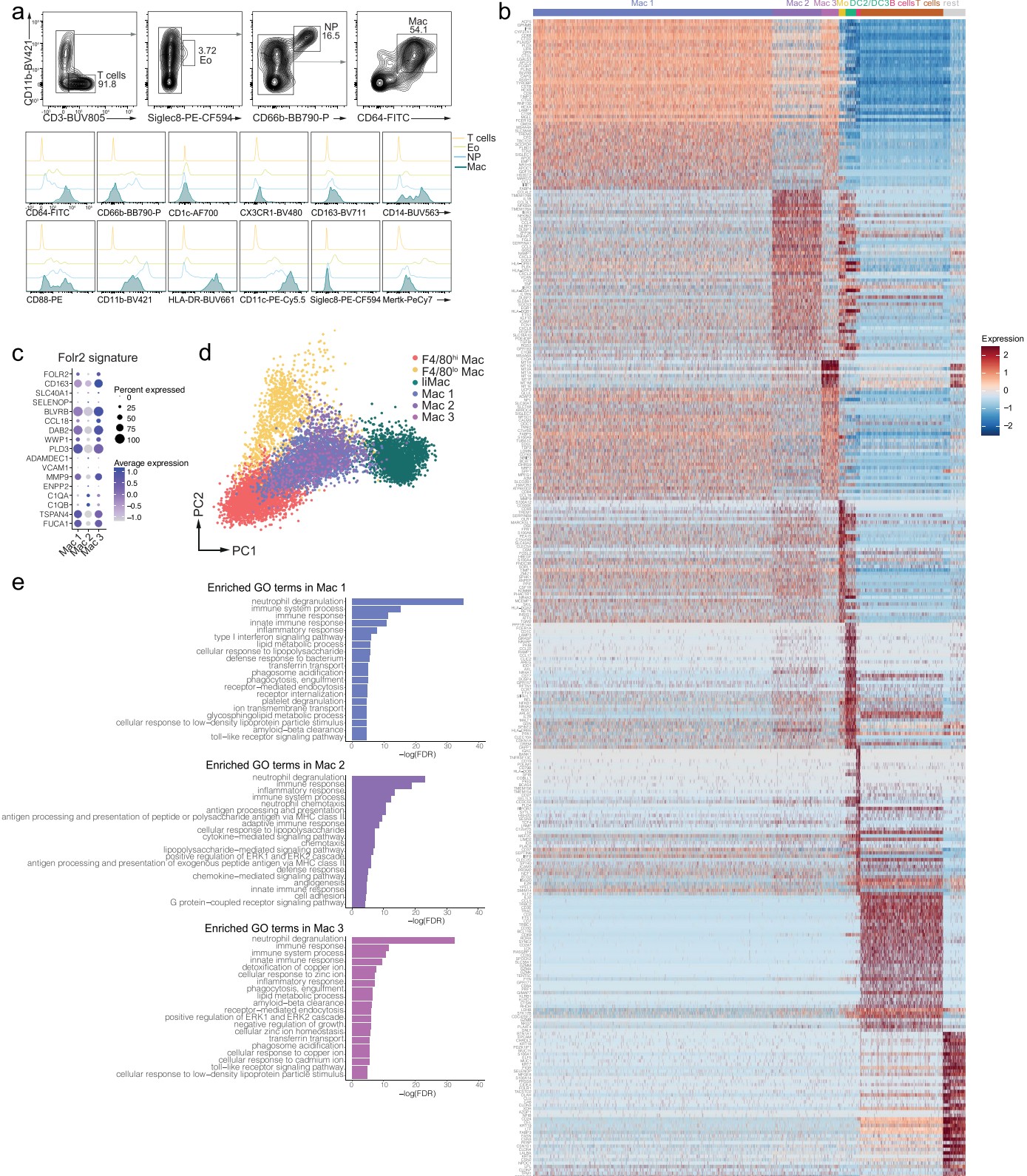

**Extended Data Fig. 7 | Three subsets of macrophages are present in human milk. a**, Representative flow cytometry plots of human breast milk T cells, eosinophils, neutrophils and macrophages (pre-gated on CD45+ cells) and representative histograms for the different markers for each population. **b**, Heatmap showing cluster-defining genes of CD45+ cells sorted from human milk samples (n = 6). Related to Fig. 6. **c**, Dot plot showing FOLR2+ macrophage

signature gene expression in Mac 1, Mac 2 and Mac 3[28]. **d**, Principal component analysis of F4/80[lo] macrophages, F4/80[hi] macrophages and liMacs of murine mammary gland (day 7 pp) and Mac 1, Mac 2 and Mac 3 populations of human milk (day 8–63 pp). **e**, Enrichment of GO terms according to the gene expression pattern in human milk Mac1, Mac2 and Mac3 as compared to T cells. Related to Fig. 6.

# nature research

# Reporting Summary

Nature Research wishes to improve the reproducibility of the work that we publish. This form provides structure for consistency and transparency in reporting. For further information on Nature Research policies, see our Editorial Policies and the Editorial Policy Checklist.

## Statistics

For all statistical analyses, confirm that the following items are present in the figure legend, table legend, main text, or Methods section.

| n/a | Confirmed | |
|---|---|---|
| ☐ | ☒ | The exact sample size (*n*) for each experimental group/condition, given as a discrete number and unit of measurement |
| ☐ | ☒ | A statement on whether measurements were taken from distinct samples or whether the same sample was measured repeatedly |
| ☐ | ☒ | The statistical test(s) used AND whether they are one- or two-sided *Only common tests should be described solely by name; describe more complex techniques in the Methods section.* |
| ☒ | ☐ | A description of all covariates tested |
| ☐ | ☒ | A description of any assumptions or corrections, such as tests of normality and adjustment for multiple comparisons |
| ☐ | ☒ | A full description of the statistical parameters including central tendency (e.g. means) or other basic estimates (e.g. regression coefficient) AND variation (e.g. standard deviation) or associated estimates of uncertainty (e.g. confidence intervals) |
| ☐ | ☒ | For null hypothesis testing, the test statistic (e.g. *F*, *t*, *r*) with confidence intervals, effect sizes, degrees of freedom and *P* value noted *Give P values as exact values whenever suitable.* |
| ☒ | ☐ | For Bayesian analysis, information on the choice of priors and Markov chain Monte Carlo settings |
| ☒ | ☐ | For hierarchical and complex designs, identification of the appropriate level for tests and full reporting of outcomes |
| ☒ | ☐ | Estimates of effect sizes (e.g. Cohen's *d*, Pearson's *r*), indicating how they were calculated |

*Our web collection on statistics for biologists contains articles on many of the points above.*

## Software and code

Policy information about availability of computer code

| Data collection | Flow cytometry data were collected using FACS Diva Software v9.1 or SpectroFlo® Software version 3.1.0. Microscopy images were acquired using LAS X software. |
|---|---|
| Data analysis | FlowJo 10.6.2 and 10.8.1 (Tree star) was used for the Flow cytometry data analysis; R and R studio v3.2-v4.0 was used for bioinformatics analysis; UMAP and FlowSOM algorithms were used from publicly available sources, which are cited in the paper; GraphPad Prism v 7.03 and 9.4.0 (Graphpad Software Inc.) was used to prepare graphs and perform statistical analysis; LAS X (Leica) and Imaris (Bitplane) software were used for image analysis; Seurat v4.1 in conjunction with Platypus v3.4.1 for scRNAseq general analysis, plots and Tabula Muris data preparation and annotation transfer; Symphony v0.1.0 for integration with Tabula Muris; Limma, edgeR and 3dVolcano for polar volcano. The analysis code has been uploaded to https://github.com/gustaveroussy/FG-Lab. |

For manuscripts utilizing custom algorithms or software that are central to the research but not yet described in published literature, software must be made available to editors and reviewers. We strongly encourage code deposition in a community repository (e.g. GitHub). See the Nature Research guidelines for submitting code & software for further information.

## Data

Policy information about availability of data

All manuscripts must include a data availability statement. This statement should provide the following information, where applicable:
- Accession codes, unique identifiers, or web links for publicly available datasets
- A list of figures that have associated raw data
- A description of any restrictions on data availability

Publicly available datasets: GRCm38.p6 Release M23, GRCh38.p13, Ensembl database, MoMAC-VERSE.
The mouse mammary gland and human milk scRNA-seq datasets are deposited in the Genome Expression Omnibus under the accession numbers: GSE230697 and GSE230749, respectively. The mass spectrometry proteomics data have been deposited to the ProteomeXchange Consortium via the PRIDE partner repository with the dataset identifier PXD041711.

# Field-specific reporting

Please select the one below that is the best fit for your research. If you are not sure, read the appropriate sections before making your selection.

☒ Life sciences        ☐ Behavioural & social sciences        ☐ Ecological, evolutionary & environmental sciences

For a reference copy of the document with all sections, see nature.com/documents/nr-reporting-summary-flat.pdf

# Life sciences study design

All studies must disclose on these points even when the disclosure is negative.

| | |
|---|---|
| Sample size | The sample size was chosen according to 3R principles, and normally included 3-5 mice per group, which is usually enough to get a statistical difference should there be any. No statistical method to predetermine sample size was used. Details on sample sizes are provided in figure legends and methods section. |
| Data exclusions | No animals were excluded from analysis unless re-genotyping revealed a wrongly determined genotype: then the animal was re-allocated in a correct group. FACS data were pregated for certain markers as described in figure legends. For scRNAseq data analysis, low quality cells exclusion and doublets filtering was done as described in methods section. |
| Replication | All experiments were independently repeated at least once and the number of experiments was stated in the figure legends. Single cell RNA sequencing analysis was performed once. |
| Randomization | No special method of randomization was used. The mice were randomly allocated to cages upon arrival from vendour or weaning. Covariates influence was controlled by keeping mice in the same conditions and repeating experiments several times. Experimental and control groups were distributed to equal numbers. |
| Blinding | It was not possible to make the investigators completely blinded to group allocation, since usually the same experimenter performed genotyping, experimental procedures, tissue harvest and data analysis, however, whenever possible, experimenters tried not to pay attention to experimental group identity before the final steps of data analysis. |

# Reporting for specific materials, systems and methods

We require information from authors about some types of materials, experimental systems and methods used in many studies. Here, indicate whether each material, system or method listed is relevant to your study. If you are not sure if a list item applies to your research, read the appropriate section before selecting a response.

### Materials & experimental systems

| n/a | Involved in the study |
|---|---|
| ☐ | ☒ Antibodies |
| ☒ | ☐ Eukaryotic cell lines |
| ☒ | ☐ Palaeontology and archaeology |
| ☐ | ☒ Animals and other organisms |
| ☐ | ☒ Human research participants |
| ☒ | ☐ Clinical data |
| ☒ | ☐ Dual use research of concern |

### Methods

| n/a | Involved in the study |
|---|---|
| ☒ | ☐ ChIP-seq |
| ☐ | ☒ Flow cytometry |
| ☒ | ☐ MRI-based neuroimaging |

## Antibodies

| | |
|---|---|
| Antibodies used | For flow cytometry on murine cells, the following antibodies were used: I-A/I-E (BioLegend art. 107620 and BD art. 746197, clone M5/114.15.2), CD11b (BD art. 612800 clone M1/70), CD11c (eBioscience art. 35-0114-82  and BioLegend art. 117331, clone N418), |

CD45 (BD art. 564279 clone 30-F11), Ly6C (BioLegend art. 128037 and art. 128023 clone HK1.4 and BD art. 553104 clone AL-21), Ly6G (BD art. 565707 clone 1A8), Siglec-F (BD art. 5626817 clone E50-2440), CD3 (Biolegend art. 100236 clone 17A2), NK1.1 (Biolegend art. 108745 clone PK136), CD19 (BD art. 612971 clone 1D3), CD64 (Biolegend art. 139309 clone X54-5/7.1), F4/80 (Biolegend art. 123112, art. 123135 clone BM8), CD169 (eBioscience art. 12-5755-80 clone SER-4), MerTK (eBioscience art. 25-5751-82 clone DS5MMER), CD206 (Biolegend art. 141734 clone C068C2), Lyve1 (eBioscience art. 50-0443-82 clone ALY7), CD38 (Biolegend art. 102714 clone 90), CX3CR1 (Biolegend art. 149027 and art. 149029 clone SA011F11), Dectin-1 (Biolegend art. 144303 clone RH1 and Biorad art. MCA2289A647T clone 2A11), XCR1 (BioLegend art. 148220 clone ZET), TNF (Biolegend art. 506328 clone MP6-XT22), and proIL-1b (eBioscience art. 25-7114-80 clone NJTEN3). Prior to detailed analysis cells were always gated on single and live cells. Dead cells were excluded with the Zombie NIR Fixable Viability Kit (Biolegend art. 423106). Click-iT EdU Cell Proliferation Kit (ThermoFisher) was used for the cell proliferation assay.

For flow cytometry on human cells CD11b (Biolegend art. 301324 clone ICRF44), CD11c (ThermoScientific art. MHCD11c18 clone BU15), CD14 (BD art. 741441 clone M5E2), CD16 (Biolegend art. 302020 clone 3G8), CD64 (Biolegend art. 305005 clone 10.1), CD66b (BD customized clone G10F5), CD3 (BD art. 612895 clone UCHT1), CD4 (BD art. 300508 clone RPAT4), CD8 (Life Technologies art. MHCD0829 clone 3B5), Siglec8 (Biolgened art. 347109 clone 7C9), MerTK (Biolegend art. 367609 clone 590H11G1E3), FceR1 (BD art. 747782 clone CRA1), CD163 (Biolegend art. 333629 clone GHI/61), CX3CR1 (BD art. 746723 clone 2A91), HLA-DR (BD art. 565073 clone G46-6), CD56 (BD art. 612766 clone NCAM16.2).

For histology, the following antibodies were used: anti-smooth muscle actin (Dako art. M0851 clone 1A4), anti-CD11c conjugated to AlexaFluor 594 (BioLegend art. 117346 clone N418), anti-Lyve1 conjugated to eFluor 660 (eBioscience art. 50-0443-82 clone ALY7), anti-Ki67 conjugated to PE-Cy7 (eBioscience art. 25-5698-82 clone SOLA15), anti-Iba1 (Wako art. 011-27991), anti-GFP (Nacalai Tesque art. 04404-84), anti-SMA (Sigma art. F3777 clone 1A4), anti-CD64 (Bio-Rad art. MCA5997 clone AT152-9), anti-Dectin-1 (Bio-Rad), anti-CD138 (BioLegend clone 281-2), anti-IgA (Southern Biotech art. 1040-01), anti-MHCII (BioLegend art. 107610 clone M5/114.15.2 (M5/114).

Anti-CSF-1 neutralizing antibody (clone 5A1) and isotype control antibody (rat IgG1, clone HPRN) were purchased from Bio X Cell (West Lebanon, NH, USA) and were administered i.p. at a dose of 0.2-0.3 mg.

| Validation | All antibodies used in this study have been previously validated by their manifacturers (the details can be checked on the manufacturers' websites by catalogue number), and further titrated and compared with FMO (FACS antibodies) in our lab. |

## Animals and other organisms

Policy information about studies involving animals; ARRIVE guidelines recommended for reporting animal research

| Laboratory animals | Female mice were normally used at the age of 7-12 weeks. The following strains were used: C57BL/6Rj, Csf1rfl/fl, ItgaxCre, Ms4a3Cre, Ccr2CreER-mKate, Il34LacZ, Ccr2-/-, Cx3cr1GFP, R26Ai14. |
| Wild animals | No wild animals were used in the study. |
| Field-collected samples | No field-collected samples were used in the study. |
| Ethics oversight | All experimental procedures at the University of Zurich were performed in accordance with Swiss Federal regulations and approved by the Cantonal Veterinary Office of Zurich. |

Note that full information on the approval of the study protocol must also be provided in the manuscript.

## Human research participants

Policy information about studies involving human research participants

| Population characteristics | Human milk samples were collected at the University Hospital Zurich from healthy women, aged 20 to 40 years [mean (SD) 32.0 (5.0)], within 4 to 63 days after delivery. |
| Recruitment | The recruitment of the study participants and the collection of human milk samples took place at the Department of Neonatology of the University Hospital Zurich. Women eligible as study participants were contacted by the project leader, informed about the present study protocol and asked for consent for participating to the study. The informed consent process included ample time for consideration given to the participants and opportunity to ask questions. No compensation or payments were given to the study participants. There were no selection biases. The milk donors could withdraw their consent at any point in time without justification. |
| Ethics oversight | The Federal Act on Research involving Human Beings (HRA, RS 810.30) and the Ordinance on Human research with the exception of clinical trials (HRO, RS 810.301, Art. 6-23). The study protocol was approved by the Swiss Ethics Commission of the Canton of Zurich (BASEC-Nr. 2020-00542) and all the subjects participating to the study signed an inform consent before the enrollment. |

Note that full information on the approval of the study protocol must also be provided in the manuscript.

# Flow Cytometry

## Plots

Confirm that:

☒ The axis labels state the marker and fluorochrome used (e.g. CD4-FITC).

☒ The axis scales are clearly visible. Include numbers along axes only for bottom left plot of group (a 'group' is an analysis of identical markers).

☒ All plots are contour plots with outliers or pseudocolor plots.

☒ A numerical value for number of cells or percentage (with statistics) is provided.

## Methodology

| | |
|---|---|
| Sample preparation | Mice were sacrificed by CO2 inhalation and intracardially perfused with phosphate buffered saline (PBS; pH 7.4, Gibco). After perfusion, left and right abdominal mammary glands were dissected, and inguinal lymph nodes removed. All the samples were cut into small pieces in an Eppendorf tube, followed by digestion in 0.4 mg/ml collagenase type IV (Worthington) and 0.04 mg/ml DNase I in Ca2+/Mg2+ HBSS supplemented with 10% FCS for 40 mins at 37°C, while shaking. Samples were homogenized with a 18 G needle and a syringe and filtered through a 100 μm cell strainer to obtain a homogeneous cell suspension. Cells were once washed in PBS, resuspended in red blood cell lysis (0.16 M NH4Cl, 0.11 M KHCO3, 0.001 M EDTA in ddH2O) and incubated on ice for 5 min, then filtered through 100 μm cell strainer and washed with PBS. <br><br> For isolation of milk cells, milk was diluted in a 1:1 ratio with PBS and centrifuged at 800g for 20 minutes at 15°C. The lipid layer and skim milk were removed, and the cell pellet was washed twice in PBS by centrifugation at 400g for 5 minutes and resuspension in PBS. <br><br> For histology, mice were euthanized through CO2 asphyxiation and perfused with PBS. Mammary fat pads were removed, fixed in 4% PFA (Morphisto) for 6-24 hours at 4°C, washed in PBS followed by incubation in 30% sucrose in PBS at 4°C for 1 to 5 days. The tissue was then embedded in Cryo Embedding Medium (Medite) and frozen on dry ice. |
| Instrument | Flow cytometry analysis was performed on LSR II Fortessa, BD FACSymphony and Cytek Aurora. Samples for scRNAseq were sorted on Aria III and S6 cell sorters. Sequencing was performed on NovaSeq6000 platform. Imaging was performed on Leica SP8 Falcon and Leica SP5 microscopes. |
| Software | FlowJo 10.6.2 and 10.8.1 (Tree star), BD FACS DIVA and  SpectroFlo® Software, Seurat v3 and v4, LAS X, Imaris |
| Cell population abundance | Frequencies and cell counts per mammary gland are specified in the figure legends. |
| Gating strategy | In general, cells were gated based on FSC-A and SSC-A to exclude debris, doublets were excluded by FSC-Area vs. FSC-Height gating. Dead cells were excluded from the analysis using Zombie NIR fixable staining reagent (BioLegend). Supplemental gating strategies are provided in Extended Data. |

☒ Tick this box to confirm that a figure exemplifying the gating strategy is provided in the Supplementary Information.

