## [Peer Review File · Nature Immunology]

Peer Review Information

Journal: Nature Immunology

Manuscript Title: Lactation-associated macrophages exist in murine mammary tissue and human milk

Corresponding author name(s): Professor Melanie Greter

Reviewer Comments & Decisions:

Decision Letter, initial version:
--

11th Jun 2021

Dear Dr. Greter,

Thank you for your response to the reviewers' comments on your Article, "A distinct lactation-associated macrophage population exists in murine mammary tissue and human milk". While they find your work of considerable potential interest, they have raised quite substantial concerns that must be addressed. In light of these comments, we cannot accept the manuscript for publication, but would be very interested in considering a revised version that addresses the referees' concerns.

Please revise along the lines specified in your letter. In addition to the model of experimental mastitis that you propose, we consider it would probably be informative to assess the role of the LiMas in protecting the pups from an infectious challenge (maybe a GI tract infection?), considering the maternal-derived macrophages can be found in the milk, so they could play a role in enhancing immunity in the pups.

At resubmission, please include a "Response to referees" detailing, point-by-point, how you addressed each referee comment. If no action was taken to address a point, you must provide a compelling argument. This response will be sent back to the referees along with the revised manuscript.

Please include a revised version of any required reporting checklist. It will be available to referees to aid in their evaluation.

The Reporting Summary can be found here:

When submitting the revised version of your manuscript, please pay close attention to our

<https://www.nature.com/nature-research/editorial-policies/image-integrity>> Digital Image Integrity Guidelines. and to the following points below:

[REDACTED]

We hope to receive the revised manuscript within 6 months. If you cannot send it within this time, please let us know. We will be happy to consider your revision so long as nothing similar has been accepted for publication at Nature Immunology or published elsewhere.

Nature Immunology is committed to improving transparency in authorship. As part of our efforts in this direction, we are now requesting that all authors identified as 'corresponding author' on published papers create and link their Open Researcher and Contributor Identifier (ORCID) with their account on the Manuscript Tracking System (MTS), prior to acceptance. ORCID helps the scientific community achieve unambiguous attribution of all scholarly contributions. You can create and link your ORCID from the home page of the MTS by clicking on 'Modify my Springer Nature account'. For more information please visit www.springernature.com/orcid.

Thank you for the opportunity to review your work.

Sincerely,

Ioana Visan, Ph.D.
Senior Editor
Nature Immunology

Tel: 212-726-9207
Fax: 212-696-9752
www.nature.com/ni

Reviewers' Comments:

Reviewer #1:

Remarks to the Author:

Summary and overall comments:

Macrophages residing in the mammary gland have been studied in a variety of contexts, including their supportive roles in ductal development, in alveolar expansion and in post-lactation involution – however, their role during lactation remains unexplored. In this study, using high dimensional flow cytometry and single-cell RNA sequencing, the authors investigated the origins, location, and maintenance of a unique population of macrophages during lactation, termed lactation-induced macrophages (LiMas). CD11chi LiMas are monocyte-derived and emerge in the lactating mammary gland 1-day post-partum. LiMas are developed/maintained in a CSF-1 and microbiota-dependent manner. This population is transcriptionally distinct from co-existing F4/80lo and F4/80hi macrophages, and numerically expand during lactation independently from monocytes. LiMas are localized to the site of milk production, extravasate into the milk during lactation, and interestingly are also present in human milk.

Overall, this paper shows for the first time a unique population of macrophages that emerges specifically during lactation and is of interest to immunologists, particularly macrophage biologists. The authors do an excellent job in investigating when the population arises, where it originates from, how it's maintained and where it's localized. The data focused on characterization was robust, however little functional relevance is shown. LiMas were not involved in mammary gland remodeling, milk production or milk composition and therefore whether they have any significant functions in lactation is not revealed. In addition, a more clear characterization of the LiMas is required. As such, the study is both interesting, and preliminary in some respects. More specific comments are detailed below.

Major comments:

1. It is appreciated the authors attempted to find specific functions for LiMas, but were not successful. To complete these experiments, and being in line with their hypothesis this subset maybe antimicrobial – some preliminary experiments to identify microbes (that commonly infect lactating tissues) and microbial ligands (TLR4, TLR2, TLR7, etc.) in vitro would be important. Do the different macrophage subpopulations respond similarly or differentially? Are the LiMas found in milk different than those still in lactating tissue? Even if similar, this would be an important finding.
2. The authors show a variety of cell surface markers for LiMas in Figure 1, and show a time course. The cells seem to be completely absent and dramatically increase in number. A more detailed time course AND corresponding cell surface marker expression would be helpful to see if they are changing marker expression. Do they express Ly6C or CCR2 in the early phases? Do they express CD11b, and downregulate them? The true source of the subtype does not seem to be known.
3. The authors assessed the localization of LiMas within the mammary gland by immunohistochemistry using Cx3cr1GFP/+ mice (Figure 1D). This data is preceded by flow cytometry using a limited number of markers. Here LiMas were identified using CX3CR1 (GFP+), CD11c and MHC-II. How can we rule out that this identified population does not represent monocytes or a recruited DC subtype, that is initially recruited, then dramatically increases in cell number? In particular, Ly6Clo monocytes may express CX3CR1, CD11c, as well as MHC-II, as can DCs, and can lack CCR2 –which would not be labeled in the CCR2-fate mapping system. Have the authors confirmed F4/80 or other macrophage marker expression in this population? CD26 can be used a classic DC marker. Additional clarifying experiments using canonical markers would be helpful here, linking flow cytometry and localization

studies. For example, sm-FISH using Flt3, CD26 (DC markers), and clear macrophage markers. Some of these data would come from the scRNA-seq studies.

4. In the scRNA-seq analysis (Figure 2), it is clear that the LiMas cluster away from the other cells. It is critically important to list a robust set of macrophage, DC (various subtypes described) markers to see the lineage of the cells. Do the LiMas express key pan macrophage genes (C1qa, C1qb), do they lack DC genes (Itgae, Dpp4, Flt3, etc.)? These data should be added.

5. The analysis to other macrophage subtypes that were obtained from bulk RNA-seq is not helpful. The authors should use published single cell analyses for such comparisons, for which there are many. The authors may use the package SingleR (or others) to manually annotate and confirm their populations, which can use a reference annotation from prior scRNA-seq studies of various macrophage subtypes.

6. LiMas are derived predominately from monocytes as shown in Figure 3. However, 25% of the population arises from another source? Are these subpopulations different, or identical? Given the focus of the paper is the characterization, it would be important to see if there are any difference. Do they localize to different areas, do they express different transcriptomes?

7. The scRNA-seq UMAP visualization data in Fig 2 suggests the LiMas may have heterogeneity. Have the author subclustered and looked for differences?

8. It is not clear why the authors specifically chose 7 days post-partum (dpp) as the timepoint for transcriptional assessment. LiMas reach their peak in the mammary gland at 12 dpp. While the number may be trivial, perhaps the abundance of LiMas present at the time may be indicative of a potential functional role, and the various pathways mediated by these LiMas may be assessed in an unbiased fashion by scRNA-seq. The authors should clarify the reasoning for their timepoint choice in the results.

9. The human scRNA-seq data should also demonstrate key macrophage genes and monocyte genes. For example, FCN1 is a monocyte gene, and C1QA, C1QB as macrophage genes. Can the authors use a more formal comparison between the human milk macrophage subsets and the murine milk subsets, such as by using the Garnett machine-learning algorithm which trains a classifier based on one dataset (mouse) and classifies cells in another (human)?

Minor comments:

1. LiMas is not an intuitive short-form for lactation-induced MACrophages – perhaps consider using “LiMacs” or “LI-Macs”.
2. Fig 2B – was it adjusted P values for single cell RNA seq data?
3. Line 70: Figure 2e doesn't seem to be relevant to this statement. Figure 2f shows enriched pathways, including TGFβ-R signaling, so this panel would be more appropriate to reference here.
4. The methods section for single-cell RNA sequencing is lacking some details. (1) Was the experiment performed once or does the data represent replicates? (2) The pathway databases that were used for the GO analysis are not specified.
5. Figure 5d – The graph on the right side is missing a y-axis label (weight)
6. Figure 6 – It would be clearer to state in the title which UMAP is from flow cytometric data versus scRNA-seq data to better distinguish between panels 6A-B versus 6C-D.

Reviewer #2:

Remarks to the Author:

The study by Cansever et al investigates the composition, origin and regulation of macrophages within the murine mammary gland post-partum and in murine milk and human breast milk, using a

combination of scRNAseq, flow-cytometry, immunofluorescence microscopy and eloquent genetic fate-mapping systems. They show that pregnancy and lactation induces massive expansion of a rare population of CSF1-dependent CD11c+ macrophages that are found within or adjacent to the milk producing alveoli. They term these Lactation-induced Macrophages (LiMas). They show LiMas undergo significant proliferation during pregnancy, and using a Ccr2-inducible cre fate mapping system they are able to show convincingly that the expansion of these cells that occurs during the lactation phase is almost exclusively due to proliferation rather than monocyte recruitment.

Although the analysis by scRNAseq is novel, these data largely corroborate and build upon recently findings published by Dawson et al. 2020, who also showed that CD11c+ ductular macrophages proliferate and massively expand in number during pregnancy and peak during lactation. Dawson et al. had also previously used CD11c-DTR mice to show that LiMas are essential for remodelling of the alveoli following weaning.

A more novel aspect of the current study is that LiMas form the predominant myeloid population present within maternal milk and that these present within milk in the stomachs of feeding pups but not within the pup's tissues. Using scRNAseq, they also identify macrophages with shared transcriptional features of murine LiMas are present within human breast milk. Although it is established that breast milk contains myeloid cells and lymphocytes (as referred to by the authors), these data refine our knowledge of those myeloid cells that are present. Finally, Cansever et al were able to deplete LiMas and leave other mammary macrophages/dendritic cells largely intact by targeting Csf1r-deficiency using CD11c-cre. However, they did not find any significant effect on milk production/composition, antibody content or pup survival.

Hence, this study is very well performed, the subject matter interesting, and a number of novel scRNAseq dataset are presented. However, my enthusiasm is somewhat tempered by the lack of a clear functional importance of these cells (beyond that reported by Dawson et al.) and the fact neither the expansion of these in the mammary gland during lactation nor the presence of myeloid cells in human/murine breast milk is wholly novel.

Have the authors looked at whether the ablation of LiMas effects the composition and number of maternal leukocytes in the milk? More importantly, they hypothesize that LiMas may be important for pathogen clearance in mastitis – can they provide evidence of this?

Other comments:

- In many places and figures the authors refer to non-lactating controls – they need to be clear what they mean by this: are these virgin mice, or pregnant mice, and is this the same in all experiments?

- No data or method is provided that support the assigned identity of the subsets identified by scRNAseq in Fig2a. Either xl sheets of the enriched genes or heat map of key marker genes.

- The transcriptional profiling in Fig 2c suggests F4/80lo mammary macrophages have a profile remarkably similar to what would be expected for classical monocytes (eg Gm9733, Chil3, Ly6c2, S100a4, Ccr2). Consistent with Ccr2 expression, the F4/80lo macrophages label well in the inducible

Ccr2-cre mice, but oddly (given their monocyte-like profile) these cells label poorly with the Ms4a3 constitutive Cre reporter in both 'nonlactating' and lactating mice. What is the reason for this – are the authors certain these do not represent monocyte-derived macrophages and that this is an oddity of the Ms4a3cre system? I note that in Line 99, the extended data Fig4a demonstrating robust labelling in monocytes, neutrophils and eosinophils in Ms4a3cre mice is referred to in text as from lactating mammary glands at 10 dpp, but the figure refers to 'non-lactating mice'. Do monocytes/neutrophils and eosinophils in lactating mice definitely label well?

- Line 116 concludes that the fate mapping data generated using the Ccr2-inducible cre mice suggests that expansion of F4/80hi macs is independent of monocytes. However, labelling of these cells in the Ms4a3 constitutive cre reporter goes up substantially between non-lactating and lactating mice suggesting monocytes are an important source of these cells. Critically, the increase in number of F4/80hi macrophages appears to occur during pregnancy and is complete by 1 dpp (then maintained steadily thereafter), thus before the labelling of monocytes was performed in the Ccr2-inducible cre mice (ie tamoxifen given at 1-3 dpp). So, it seems more likely that F4/80hi macrophages expand in number between pregnancy and 1dpp at least in part by differentiation of monocytes but then are maintained by self-renewal during lactation – this should be discussed

Author Rebuttal to Initial comments
--

See Inserted PDF

Point-by-point (PbP) response

Reviewer #1

Summary and overall comments:

Macrophages residing in the mammary gland have been studied in a variety of contexts, including their supportive roles in ductal development, in alveolar expansion and in post-lactation involution – however, their role during lactation remains unexplored. In this study, using high dimensional flow cytometry and single-cell RNA sequencing, the authors investigated the origins, location, and maintenance of a unique population of macrophages during lactation, termed lactation-induced macrophages (LiMas). CD11chi LiMas are monocyte-derived and emerge in the lactating mammary gland 1-day post-partum. LiMas are developed/maintained in a CSF-1 and microbiota-dependent manner. This population is transcriptionally distinct from co-existing F4/80^{lo} and F4/80^{hi} macrophages, and numerically expand during lactation independently from monocytes. LiMas are localized to the site of milk production, extravasate into the milk during lactation, and interestingly are also present in human milk.

Overall, this paper shows for the first time a unique population of macrophages that emerges specifically during lactation and is of interest to immunologists, particularly macrophage biologists. The authors do an excellent job in investigating when the population arises, where it originates from, how it's maintained and where it's localized. The data focused on characterization was robust, however little functional relevance is shown. LiMas were not involved in mammary gland remodeling, milk production or milk composition and therefore whether they have any significant functions in lactation is not revealed. In addition, a more clear characterization of the LiMas is required. As such, the study is both interesting, and preliminary in some respects. More specific comments are detailed below.

We thank the reviewer for the positive comments and the astute summary.

Major comments:

1. It is appreciated the authors attempted to find specific functions for LiMas, but were not successful. To complete these experiments, and being in line with their hypothesis this subset maybe antimicrobial – some preliminary experiments to identify microbes (that commonly infect lactating tissues) and microbial ligands (TLR4, TLR2, TLR7, etc.) *in vitro* would be important. Do the different macrophage subpopulations respond similarly or differentially?

This is a good suggestion. To address the antimicrobial function of the different mammary gland macrophage populations, we stimulated mammary gland macrophages with TLR2 or TLR4 ligands (LPS and Zymosan, respectively) *in vitro*. LiMacs had the highest expression of pro-IL1 β in both conditions and interestingly, already in the unstimulated samples, LiMacs expressed pro-IL1 β , in agreement with the scRNA-seq data (new Extended Data Fig. 6a-b, Fig. 2c). Conversely, while LiMacs displayed increased TNF production upon Zymosan and LPS treatment compared to untreated conditions, their levels appeared to be consistently lower than in F4/80^{hi} macrophages (new Extended Data Fig. 6a-b).

We next assessed the phagocytic capacity of mammary gland macrophages from the lactating mammary gland by exposing them to pH-sensitive pHrodoTM Red-Zymosan, pHrodoTM Red-*E. coli* or pHrodoTM Red-*S. aureus* bioparticles. LiMacs were superior in taking up *E. coli* particles compared to F4/80^{hi} and F4/80^{lo} macrophages, whereas F4/80^{lo} macrophages displayed increased internalization of Zymosan particles. No significant difference in particle uptake was found between the three mammary gland macrophages upon exposure to *S. aureus* particles (new Extended Data Fig. 6c-d).

Altogether, these data demonstrate that LiMacs are phagocytic and responsive to microbial stimuli.

Lastly, to interrogate the potential role of LiMacs in an inflamed mammary gland *in vivo*, we used a model of experimental mastitis induced by LPS injection into the mammary gland of lactating mice. This resulted in infiltration of immune cells into alveolar spaces 18 hours after LPS challenge (new Fig. 5f), characterized mostly by invading neutrophils (new Fig. 5g-h, new Extended Data Fig. 6e). In the absence of LiMacs (α -CSF-1 antibody-treated lactating WT mice), a significant reduction of neutrophils was detected in the LPS-injected mammary gland (new Fig. 5h, new Extended Data Fig. 6e).

Altogether, these data suggest that LiMacs are equipped to respond to microbial stimulation and are activated upon LPS challenge *in vivo*, leading to the recruitment of neutrophils, which were previously shown to be critical for resolving *E. coli*-induced mastitis.

2. Are the LiMas found in milk different than those still in lactating tissue? Even if similar, this would be an important finding.

This is an interesting point. We aimed to perform scRNA-seq of immune cells derived from murine milk. However, due to the extremely low cell number of CD45⁺ cells in the milk (less than 1%) and the low volume/yield of murine milk, we did unfortunately not achieve enough reads for LiMacs. Instead, we analyzed CD45⁺ cells in the milk by flow cytometry and found that they were also positive for F4/80, CX3CR1, MHCII, Dectin-1 and negative for CD11b, similar to the LiMacs in the lactating mammary gland (new Fig. 5b-c).

Of note, mammary gland single cell suspensions (for flow cytometry and scRNA-seq) contain LiMacs in the tissue and LiMacs in the milk-filled lumen of the alveoli. Yet, apart from the small cluster of proliferating LiMacs (new Fig. 2d-e, Extended Data Fig. 4a), neither scRNA-seq at different time points nor flow cytometry revealed heterogeneity within LiMacs, suggesting that they are phenotypically similar in the mammary gland and the milk.

3. The authors show a variety of cell surface markers for LiMacs in Figure 1, and show a time course. The cells seem to be completely absent and dramatically increase in number. A more detailed time course AND corresponding cell surface marker expression would be helpful to see if they are changing marker expression. Do they express Ly6C or CCR2 in the early phases? Do they express CD11b, and downregulate them? The true source of the subtype does not seem to be known.

This is an important comment. Throughout the manuscript, LiMacs were defined as Ly6C⁻CD11b⁻ and they were also negative for *Ccr2* according to our scRNA-seq (new Extended Data Fig. 3a) and flow cytometry data (PbP Fig. 2). The fate-mapping analyses (Fig. 3) demonstrated that LiMacs emerge initially from Ly6C^{hi} monocytes and accumulate *in situ* during lactation. Even at the early time point 1 dpp, we did not detect a precursor population (for example expressing Ly6C or CD11b at intermediate levels) differentiating into LiMacs (PbP Fig. 1, below).

To more extensively characterize LiMacs over time and to analyze whether they further differentiate, we performed additional scRNA-seq of mammary glands of lactating WT mice at 8, 11, and 14 dpp (new Fig. 2d-e, new Extended Data Fig. 4b). These new data revealed that LiMacs are a relatively homogenous population, which did not further differentiate or change across the three time points analyzed. As shown for the 7 dpp time point (Extended Data Fig. 4a), a small proliferative cluster within the LiMacs was also evident at these additional three time points (8, 11, 14 dpp) (new Fig. 2d-e), further corroborating our finding that LiMacs accumulate locally without the influx of circulating monocytes (Fig. 3).

PbP Fig. 1: No intermediate precursor population can be identified early post-partum.

Flow cytometry analysis of lactating mammary glands from WT mice at 1, 6 and 15 dpp. Representative flow cytometry plots show CD11c⁺ and CD11b⁺ cells (pre-gated on CD45⁺SiglecF⁻Ly6G⁻ cells), and MHCII⁺ vs. Ly6C⁺ cells. F4/80^{hi} and F4/80^{lo} macrophages were excluded, as illustrated.

4. The authors assessed the localization of LiMas within the mammary gland by immunohistochemistry using Cx3cr1GFP/+ mice (Figure 1D). This data is preceded by flow cytometry using a limited number of markers. Here LiMas were identified using CX3CR1 (GFP+), CD11c and MHC-II. How can we rule out that this identified population does not represent monocytes or a recruited DC subtype, that is initially recruited, then dramatically increases in cell number?

In particular, Ly6Clo monocytes may express CX3CR1, CD11c, as well as MHC-II, as can DCs, and can lack CCR2 –which would not be labeled in the CCR2-fate mapping system. Have the authors confirmed F4/80 or other macrophage marker expression in this population? CD26 can be used a classic DC marker. Additional clarifying experiments using canonical markers would be helpful here, linking flow cytometry and localization studies. For example, sm-FISH using Flt3, CD26 (DC markers), and clear macrophage markers. Some of these data would come from the scRNA-seq studies.

This is an important point. While by flow cytometry and scRNA-seq different myeloid populations including monocytes, DCs and the 3 subsets of mammary gland macrophages could be identified (as demonstrated in Figs. 1, 2), for conventional immunofluorescence stainings the number of markers that can be used on a section is indeed limited. Thus, to further validate that LiMacs are *bona fide* macrophages, we added the expression of characteristic macrophage vs. DC-lineage genes obtained from the scRNA-seq data in the revised version of the manuscript (see also response to question 5 below). LiMacs expressed macrophage core genes including *Csf1r*, *Aif1*, *C1qa*, *C1qb* but were largely negative for classic cDC genes, such as *Ccr7*, *Flt3* and *Zbtb46* and *Dpp4* (CD26) (new Extended Data Fig. 3c). Of note, cDCs did not express *Cx3cr1* in the lactating mammary gland (new Extended Data Fig. 3c).

Yet, to further demonstrate that the (Cx3cr1)GFP^{hi} cells at the alveoli of lactating mammary glands of *Cx3cr1*^{GFP/+} mice correlate to LiMacs and not to cDCs, we used canonical macrophage markers such as CD64 and Iba1, for immunohistochemistry (IHC). The abundant GFP^{hi} cells at the alveoli also ubiquitously co-expressed CD64 and Iba1 (new Extended Data Fig. 2c-d). In addition, by scRNA-seq, we identified *Clec7a* (Dectin-1) as a distinctive LiMac signature gene (Fig. 2c), which also proved to be a useful marker for LiMacs by IHC. Virtually all GFP⁺ cells at the alveoli were also positive for Dectin-1 (new Extended Data Fig. 3f).

Furthermore, GFP⁺Lyve1⁺ cells and also GFP^{dim}MHCII⁺ cells residing outside alveoli as shown in the Extended Data Fig. 2a-b, represent F4/80^{hi} and F4/80^{lo} macrophages (or cDCs), respectively.

To show that (CX3CR1)GFP^{hi} cells adjacent to alveoli do not represent monocytes, we used *Ccr2*^{CreER}*R26*^{Ai14} mice. For clarification, these are fate-mapping mice and the labeling is induced by tamoxifen injections, which leads to the tagging of Ly6C^{hi} and Ly6C^{lo} monocytes and partially cDCs given that their precursors in the BM express *Ccr2*, as previously shown (Croxford et al. *Immunity* 2015, Utz et al. *Cell* 2020, Amorim et al. *Nature Immunology* 2022). In these tamoxifen-treated *Ccr2*^{CreER}*R26*^{Ai14} mice, the majority of LiMacs remained unlabeled (Fig. 3c), while monocytes were efficiently tagged in the lactating mammary gland. We further included immunofluorescence analysis, demonstrating that the majority of Iba1⁺ cells were tdTomato⁻ and adjacent to alveoli with a ramified and dendritic cell shape (new Fig. 3d).

Conversely, in lactating mammary glands of *Ms4a3*^{CreER}*R26*^{Ai14} mice, Dectin-1⁺ cells were labeled with tdTomato (new Fig. 3b), excluding them from being cDCs as these fate-mapping mice faithfully label monocytes and granulocytes but not cDCs (Liu et al. *Cell* 2019).

Altogether, with these additional experiments, we can hopefully convince the reviewer that the GFP^{hi} and CD11c⁺ / MHCII⁺ / CD64⁺ cells are macrophages (LiMacs) associated to alveoli.

5. In the scRNA-seq analysis (Figure 2), it is clear that the LiMas cluster away from the other cells. It is critically important to list a robust set of macrophage, DC (various subtypes described) markers to see the lineage of the cells. Do the LiMas express key pan macrophage genes (C1qa, C1qb), do they lack DC genes (Itgae, Dpp4, Flt3, etc.)? These data should be added.

This is a good suggestion. As mentioned above (response to comment 4), we have added cDC- and macrophage-defining genes (as originally defined in Gautier et al. *Nature Immunology* 2012, Miller et al. *Nature Immunology* 2012) in the revised version of the manuscript and included the heatmap for the

defined clusters (new Extended Data Fig. 3a, c). LiMacs express *C1qa* and *C1qb* and other 'macrophage genes' but do mostly not express cDC-defining genes including *Itgae*, *Dpp4*, *Ccr7* or *Flt3*.

6. The analysis to other macrophage subtypes that were obtained from bulk RNA-seq is not helpful. The authors should use published single cell analyses for such comparisons, for which there are many. The authors may use the package SingleR (or others) to manually annotate and confirm their populations, which can use a reference annotation from prior scRNA-seq studies of various macrophage subtypes.

We agree and have thus removed original Fig. 2e where we compared LiMacs to the transcriptome of different tissue resident macrophage populations (obtained by bulk sequencing).

Instead, we extended our analysis of the scRNA-seq data of the murine mammary gland macrophages and human milk macrophages (see also response to point 11 below): We first analyzed the similarity between the murine mammary gland macrophages in the lactating mammary gland and the 3 human milk macrophage subsets. Similarity score analysis revealed that LiMacs and murine F4/80^{lo} macrophages resembled primarily the macrophage (Mac) 2 subset of human milk macrophages, while murine F4/80^{hi} macrophages aligned closer to both human Mac 1 and Mac 2 (new Fig. 6f). Principal Component Analysis (PCA) of the 3 murine mammary gland and the 3 human milk macrophages revealed a similar trend (new Extended Data Fig. 7d).

Specifically, F4/80^{hi} macrophages were classified by *Mrc1*, *Cd163*, *Lyve1*, *Folr2*, *Mgl2* and *Pf4*, genes previously described to be expressed by tissue-resident macrophages associated to vessels, for example in the brain, the skin or also in the mammary gland (Chakarov et al. *Science* 2019, Dawson et al. *Nature Cell Biology* 2020, Jäppinen et al. *Nature Communications* 2019, Ramos et al. *Cell* 2021, Utz et al. *Cell* 2020). In the mammary gland, they relate to the recently described *Folr2*⁺ macrophages (new Extended Data Fig. 3b) (Ramos et al. *Cell* 2021).

In this report (Ramos et al. *Cell* 2021), high expression of *Cadm1* and *Olfml3* was identified on tumor-associated TREM2⁺ macrophages in the mammary gland. Here, LiMacs were not specifically enriched for other signature genes characterizing the *Cadm1*⁺ subset of breast cancer-associated macrophages (new Extended Data Fig. 3e). This suggests that LiMacs are a distinct population in lactation and are likely unrelated to tumor-associated macrophages.

Furthermore, we compared human milk macrophages to other human tissue macrophages across multiple organs in healthy and pathological conditions (MoMac-Verse as described in Mulder et al. *Immunity* 2021). Interestingly, we discovered that the majority of human milk macrophages were similar to a macrophage cluster called 'TREM2' macrophages in the 'MoMac-Verse' (new Fig. 6g-i). This TREM2 signature has been described to be enriched in tumor-associated macrophages with immunosuppressive properties and has also been found in lipid-associated macrophages (Jaitin et al. *Cell* 2019, Katzenelenbogen et al. *Cell* 2020, Mulder et al. *Immunity* 2021, Ramos et al. *Cell* 2022). Altogether, these data show that human milk also contains macrophages, which partly resemble murine LiMacs in phenotype and transcriptome, and which display a profile reminiscent of lipid- or cancer-associated macrophages.

7. LiMas are derived predominately from monocytes as shown in Figure 3. However, 25% of the population arises from another source? Are these subpopulations different, or identical? Given the focus of the paper is the characterization, it would be important to see if there are any difference. Do they localize to different areas, do they express different transcriptomes?

It was previously shown (Liu et al. *Cell* 2019) that monocytes are labeled at almost 100% in *Ms4a3*^{CreER}*R26*^{Ai14} mice, whereas well-characterized monocyte-derived gut and dermal macrophages for example were labeled at approximately 80%, reaching a plateau at 20 weeks of age. These data in the Liu et al. report correlate to the labeling of LiMacs, which was consistently around 80%, as assessed by flow cytometry (new Fig. 3a). We did not detect any difference in surface marker expression between the labeled and the unlabeled LiMacs by flow cytometry (see PbP Fig. 2, below).

Regarding the location of labeled vs. unlabeled LiMacs, new immunofluorescence stainings of lactating mammary glands of *Ms4a3*^{Cre}*R26*^{Ai14} mice revealed that virtually all Dectin-1⁺ cells at the alveoli

(representing LiMacs, new Extended Data Fig. 3f) were tdTomato⁺ (new Fig. 3b), again suggesting that they are mostly derived from monocytes.

Furthermore, we included additional scRNA-seq data of LiMacs at different time points post-partum (as described above) (new Fig. 2d-e), which did also not reveal heterogeneity in the LiMac population (see above, new Fig. 2d-e, new Extended Fig. 4b). In summary, the majority of LiMacs is originally derived from BM-monocytes and expands during lactation via proliferation.

PbP Fig. 2: tdTomato⁺ and tdTomato⁻ LiMacs are identical.

Flow cytometry analysis of LiMacs (pre-gated on CD45⁺SiglecF⁻Ly6G⁻Ly6C⁻ cells) in the lactating mammary glands of *Ms4a3^{Cre}R26^{Air14}* mice between 7 - 10 dpp. Representative histograms show the expression of the respective markers in either tdTomato(tdTom)⁺ (red) or tdTomato⁻ (green) LiMacs or as controls total CD45⁺ cells, neutrophils, F4/80^{hi} or F4/80^{lo} macrophages (in grey).

8. The scRNA-seq UMAP visualization data in Fig 2 suggests the LiMas may have heterogeneity. Have the author subclustered and looked for differences?

This is a good idea. We found that the ‘smaller’ cluster mostly represented proliferating LiMacs (see also Extended Data Fig. 4a). To further assess whether there is additional heterogeneity among LiMacs and also across time, we performed scRNA-seq of CD11b⁺/CD11c⁺ cells from the lactating mammary glands at 8, 11 and 14 dpp. Interestingly, no further subsets or additional differently expressed genes could be detected across the 3 time points (new Fig. 2d-e, new Extended Data Fig. 4b). Nevertheless, at all time points, as observed for 7 dpp, a cluster of proliferating LiMacs could be identified (new Fig. 2d-e). Thus, LiMacs are a rather homogenous population, which increases in numbers over the time of lactation.

9. It is not clear why the authors specifically chose 7 days post-partum (dpp) as the timepoint for transcriptional assessment. LiMas reach their peak in the mammary gland at 12 dpp. While the number may be trivial, perhaps the abundance of LiMas present at the time may be indicative of a potential functional role, and the various pathways mediated by these LiMas may be assessed in an unbiased fashion by scRNA-seq. The authors should clarify the reasoning for their timepoint choice in the results.

This is a good comment that needs clarification. We originally isolated myeloid cells at 7 dpp from the mammary gland at a time point where LiMacs were abundant but early enough to capture also potentially differentiating cells. To analyze whether the transcriptome of LiMacs changes over the course of lactation, we performed additional scRNA-seq of LiMacs at 8, 11, 14 dpp (new Fig. 2d-e, new Extended Data Fig. 4b). These new data demonstrate that the profile of LiMacs did not change over time and did also not reveal further heterogeneity, apart from the subset of proliferating LiMacs.

10. The human scRNA-seq data should also demonstrate key macrophage genes and monocyte genes. For example, FCN1 is a monocyte gene, and C1QA, C1QB as macrophage genes.

We agree, as suggested, we included in the revised manuscript the expression of typical macrophage (*CSF1R*, *MARCO*, *CD68*, *CD163*, *SIGLEC1*, *AIF1*, *C1QA*, *C1QB*), cDC1 (such as *XCR1*), cDC2 (*CD1C*, *CD1E*, *FCER1A*) and monocytes genes (*S100A8*, *S100A9* and *VCAN*) by the 3 subsets of human milk macrophages (new Fig. 6d).

11. Can the authors use a more formal comparison between the human milk macrophage subsets and the murine milk subsets, such as by using the Garnett machine-learning algorithm which trains a classifier based on one dataset (mouse) and classifies cells in another (human)?

This is a good idea. Unfortunately, this suggested algorithm did not reveal any meaningful comparisons with our data.

(see response to point 6 above): *Instead, we first analyzed the similarity between the murine mammary gland macrophages in the lactating mammary gland and the 3 human milk macrophage subsets. Similarity score analysis revealed that LiMacs and murine F4/80^{lo} macrophages resembled primarily the macrophage (Mac) 2 subset of human milk macrophages, while murine F4/80^{hi} macrophages aligned closer to both human Mac 1 and Mac 2 (new Fig. 6f). In addition, PCA of the 3 murine mammary gland and the 3 human milk macrophages revealed a similar trend (new Extended Data Fig. 7c).*

Furthermore, we compared human milk macrophages to other human tissue macrophages across multiple organs in healthy and pathological conditions (MoMac-Verse as described in Mulder et al. Immunity 2021). Interestingly, we discovered that the majority of human milk macrophages correlated to a macrophage cluster called 'TREM2' macrophages in the 'MoMac-Verse' (new Fig. 6g-i). This TREM2 signature has been described to be enriched in tumor-associated macrophages with immunosuppressive properties and has also been found in lipid-associated macrophages (Jaitin et al. Cell 2019, Katzenelenbogen et al. Cell 2020, Mulder et al. Immunity 2021, Ramos et al. Cell 2022). Altogether, these data show that human milk also contains macrophages, which partly resemble murine LiMacs in phenotype and transcriptome, and which display a profile reminiscent of lipid- or cancer-associated macrophages.

Minor comments:

1. LiMas is not an intuitive short-form for lactation-induced MACrophages – perhaps consider using “LiMacs” or “LI-Macs”.

This is a great suggestion. We have changed LiMas to LiMacs in the revised version of the manuscript.

2. Fig 2B – was it adjusted P values for single cell RNA seq data?

No, the Venn diagram shows genes expressed in the F480^{lo} macrophages (Mac), F480^{hi} macrophages and LiMacs.

3. Line 70: Figure 2e doesn't seem to be relevant to this statement. Figure 2f shows enriched pathways, including TGFβ-R signaling, so this panel would be more appropriate to reference here.

We corrected this.

4. The methods section for single-cell RNA sequencing is lacking some details. (1) Was the experiment performed once or does the data represent replicates? (2) The pathway databases that were used for the GO analysis are not specified.

We provided these details in the revised manuscript.

(1) For the murine data, we performed the scRNA-seq experiment for 7 dpp once, with n=2 for lactating mammary glands and n=1 for virgin mammary gland, the scRNA-seq for 8 dpp, 11 dpp, 14 dpp was also performed once, with n=3 per time point. For the human scRNA-seq data, the experiment was performed once, with 6 donors.

5. Figure 5d – The graph on the right side is missing a y-axis label (weight)

Yes, thank you, we corrected this.

6. Figure 6 – It would be clearer to state in the title which UMAP is from flow cytometric data versus scRNA-seq data to better distinguish between panels 6A-B versus 6C-D.

This is a good idea. We added titles to these panels.

Reviewer #2

The study by Cansever et al investigates the composition, origin and regulation of macrophages within the murine mammary gland post-partum and in murine milk and human breast milk, using a combination of scRNAseq, flow-cytometry, immunofluorescence microscopy and eloquent genetic fate-mapping systems. They show that pregnancy and lactation induces massive expansion of a rare population of CSF1-dependent CD11c+ macrophages that are found within or adjacent to the milk producing alveoli. They term these Lactation-induced Macrophages (LiMas). They show LiMas undergo significant proliferation during pregnancy, and using a Ccr2-inducible cre fate mapping system they are able to show convincingly that the expansion of these cells that occurs during the lactation phase is almost exclusively due to proliferation rather than monocyte recruitment.

Although the analysis by scRNAseq is novel, these data largely corroborate and build upon recently findings published by Dawson et al. 2020, who also showed that CD11c+ ductular macrophages proliferate and massively expand in number during pregnancy and peak during lactation. Dawson et al. had also previously used CD11c-DTR mice to show that LiMas are essential for remodelling of the aveoli following weaning.

A more novel aspect of the current study is that LiMas form the predominant myeloid population present within maternal milk and that these present within milk in the stomachs of feeding pups but not within the pup's tissues. Using scRNAseq, they also identify macrophages with shared transcriptional features of murine LiMas are present within human breast milk. Although it is established that breast milk contains myeloid cells and lymphocytes (as referred to by the authors), these data refine our knowledge of those myeloid cells that are present. Finally, Cansever et al were able to deplete LiMas and leave other mammary macrophages/dendritic cells largely intact by targeting Csf1r-deficiency using CD11c-cre. However, they did not find any significant effect on milk production/composition, antibody content or pup survival.

Hence, this study is very well performed, the subject matter interesting, and a number of novel scRNAseq dataset are presented. However, my enthusiasm is somewhat tempered by the lack of a clear functional importance of these cells (beyond that reported by Dawson et al.) and the fact neither the expansion of these in the mammary gland during lactation nor the presence of myeloid cells in human/murine breast milk is wholly novel.

We thank the reviewer for the positive comments.

1. Have the authors looked at whether the ablation of LiMas effects the composition and number of maternal leukocytes in the milk (A)? More importantly, they hypothesize that LiMas may be important for pathogen clearance in mastitis – can they provide evidence of this (B)?

(A) This is an important point. Analyzing the immune compartment in the milk is difficult due to the low yield of viable cells. However, analyzing total mammary gland tissue includes cells present in the mammary gland tissue as well as in the milk-filled lumen of the alveoli. We found that the overall numbers and composition of immune cells obtained from the lactating mammary gland (tissue and milk) did not significantly change in the absence of LiMacs in *Cd11c^{Cre}Csf1r^{fl/fl}* mice or in α -CSF-1 antibody treated mice compared to controls (new Extended Data Fig. 4d-e).

(B) To interrogate the potential role of LiMacs in an inflamed mammary gland in vivo, we used a model of experimental mastitis induced by LPS injection into the mammary gland of lactating mice (see also response to 1 of reviewer #1). This resulted in infiltration of immune cells into alveolar spaces 18 hours after LPS challenge (new Fig. 5f), characterized mostly by invading neutrophils (new Fig. 5g-h, Extended Data Fig. 6e). In the absence of LiMacs (α -CSF-1 antibody-treated lactating WT mice), a significant reduction of neutrophils was detected in the LPS-injected mammary gland (new Fig. 5g-h,

Extended Data Fig. 6e). These data suggest that LiMacs are activated upon LPS challenge in vivo, leading to the recruitment of neutrophils, which were previously shown to be critical for resolving *E. coli*-induced mastitis.

In addition, we also assessed the phagocytic capacity of mammary gland macrophages from the lactating mammary gland by exposing them to pH-sensitive pHrodo™ Red-Zymosan, pHrodo™ Red-*E. coli* or pHrodo™ Red-*S. aureus* bioparticles. LiMacs were superior in taking up *E. coli* particles compared to F4/80^{hi} and F4/80^{lo} macrophages, whereas F4/80^{lo} macrophages displayed increased internalization of Zymosan particles. No significant difference in particle uptake was found between the three mammary gland macrophages upon exposure to *S. aureus* particles (new *Extended Data Fig. 6c-d*). Altogether, these data demonstrate that LiMacs are phagocytic and responsive to microbial stimuli.

Other comments:

- In many places and figures the authors refer to non-lactating controls – they need to be clear what they mean by this: are these virgin mice, or pregnant mice, and is this the same in all experiments?

This is a good point. Non-lactating controls means virgin mice throughout the manuscript. We clarified this and wrote 'virgin' controls instead.

- No data or method is provided that support the assigned identity of the subsets identified by scRNAseq in Fig2a. Either xl sheets of the enriched genes or heat map of key marker genes.

This is a good point. We included a heatmap used to identify the cell populations in the scRNA-seq data (new *Extended Data Fig. 3a, c*).

- The transcriptional profiling in Fig 2c suggests F4/80^{lo} mammary macrophages have a profile remarkably similar to what would be expected for classical monocytes (eg Gm9733, Chil3, Ly6c2, S100a4, Ccr2). Consistent with Ccr2 expression, the F4/80^{lo} macrophages label well in the inducible Ccr2-cre mice, but oddly (given their monocyte-like profile) these cells label poorly with the Ms4a3 constitutive Cre reporter in both 'nonlactating' and lactating mice. What is the reason for this – are the authors certain these do not represent monocyte-derived macrophages and that this is an oddity of the Ms4a3cre system?

This is a valid comment. As pointed out by the reviewer, while some genes typically represented by monocytes were also expressed by F4/80^{lo} macrophages (for example *Ccr2*), they also expressed several 'macrophage signature genes' (new *Extended Data Fig. 3c*). Of note, their expression of *Ccr2* results in their tagging in *Ccr2*^{CreER} mice (Fig. 3c). On the other hand, in *Ms4a3*^{Cre} mice, embryonically-derived tissue resident macrophages are not labeled (Liu et al. *Cell* 2019), whereas Ly6C^{hi} and Ly6C^{lo} monocytes are efficiently tagged, which we also verified in the lactating mammary gland (PbP Fig. 3, below). Yet, F4/80^{lo} were previously shown to be embryonically-derived and gradually replaced by monocytes over time (Dawson et al. *Nature Cell Biology* 2020). In line with this, the *Ms4a3*^{Cre}*R26*^{Ai14} mice in our study were approximately 2 months old, in which the F4/80^{lo} macrophages were labeled at 35% (Fig. 3a). Thus, *Ccr2*-expressing F4/80^{lo} macrophages appear to be embryonically-derived and are likely gradually replaced by monocytes over time.

Yet, we totally agree with the reviewer that the F4/80^{lo} macrophages are an enigmatic population and perhaps heterogenous and we have discussed this in the revised manuscript (line 321).

I note that in Line 99, the extended data Fig4a demonstrating robust labelling in monocytes, neutrophils and eosinophils in Ms4a3cre mice is referred to in text as from lactating mammary glands at 10 dpp, but the figure refers to 'non-lactating mice'. Do monocytes/neutrophils and eosinophils in lactating mice definitely label well?

Extended Data Fig. 4a originally showed non-lactating mammary glands. However, for clarification, we only show now the lactating mammary glands in the revised version of the manuscript (Fig. 3a). No difference could be detected in the labeling of neutrophils, eosinophils, Ly6C^{hi} and Ly6C^{lo} monocytes between virgin and lactating mammary glands of *Ms4a3*^{Cre}*R26*^{Ai14} mice (see below, PbP Fig. 3).

PbP Fig. 3: No difference in labeling between virgin and lactating mammary glands of *Ms4a3^{Cre}R26^{Ai14}* mice. Flow cytometry analysis of neutrophils (Ly6G⁺CD11b⁺), eosinophils (SiglecF⁺CD11b⁺), Ly6C^{hi} monocytes (Mo) (Ly6C^{hi}CD11b⁺) and Ly6C^{lo} monocytes (CD11b⁺Cx3cr1⁺F4/80^{int}) (pre-gated on CD45⁺ cells) of virgin and lactating mammary glands (8-15 dpp) of *Ms4a3^{Cre}R26^{Ai14}* mice.

- Line 116 concludes that the fate mapping data generated using the *Ccr2*-inducible cre mice suggests that expansion of F4/80hi macs is independent of monocytes. However, labelling of these cells in the *Ms4a3* constitutive cre reporter goes up substantially between non-lactating and lactating mice suggesting monocytes are an important source of these cells. Critically, the increase in number of F4/80hi macrophages appears to occur during pregnancy and is complete by 1 dpp (then maintained steadily thereafter), thus before the labelling of monocytes was performed in the *Ccr2*-inducible cre mice (ie tamoxifen given at 1-3 dpp). So, it seems more likely that F4/80hi macrophages expand in number between pregnancy and 1dpp at least in part by differentiation of monocytes but then are maintained by self-renewal during lactation – this should be discussed

This is an important point and good explanation. We discussed it in more detail (line 325). In addition, we believe that the difference between the labeling in the *Ms4a3^{Cre}* mice between virgin and non-lactating mammary glands in the F4/80^{hi} macrophages is likely due to an age difference of the mice. F4/80^{hi} macrophages were previously shown to be replaced by monocytes with increasing age (Dawson et al. *Nature Cell Biology* 2020). For clarity, we removed the virgin mammary gland and only show the labeling in the lactating mammary gland where LiMacs are present (new Fig. 3a).

Decision Letter, first revision:

21st Feb 2023

Dear Dr. Greter,

Your Article, "Distinct lactation-associated macrophages exist in murine mammary tissue and human milk" has now been seen by the original referees. Although we are interested in the possibility of publishing your study in Nature Immunology, the remaining concerns of the referees need to be addressed.

Please revise to address these remaining issues. At resubmission, please include a "Response to referees" detailing, point-by-point, how you addressed each referee comment. If no action was taken to address a point, you must provide a compelling argument. This response will be sent back to the referees along with the revised manuscript.

Please include a revised version of any required reporting checklist. It will be available to referees to aid in their evaluation.

Reporting summary:

When submitting the revised version of your manuscript, please pay close attention to our [href="https://www.nature.com/nature-portfolio/editorial-policies/image-integrity">Digital Image Integrity Guidelines](https://www.nature.com/nature-portfolio/editorial-policies/image-integrity) and to the following points below:

[REDACTED]

We hope to receive your revised manuscript within three-four weeks. If you cannot send it within this time, please let us know. We will be happy to consider your revision so long as nothing similar has been accepted for publication at Nature Immunology or published elsewhere.

Please do not hesitate to contact me if you have any questions or would like to discuss these revisions

further.

Nature Immunology is committed to improving transparency in authorship. As part of our efforts in this direction, we are now requesting that all authors identified as 'corresponding author' on published papers create and link their Open Researcher and Contributor Identifier (ORCID) with their account on the Manuscript Tracking System (MTS), prior to acceptance. ORCID helps the scientific community achieve unambiguous attribution of all scholarly contributions. You can create and link your ORCID from the home page of the MTS by clicking on 'Modify my Springer Nature account'. For more information please visit www.springernature.com/orcid.

Sincerely,

Ioana Visan, Ph.D.
Senior Editor
Nature Immunology

Tel: 212-726-9207
Fax: 212-696-9752
www.nature.com/ni

Reviewers' Comments:

Reviewer #1:

Remarks to the Author:

The authors have submitted a revised manuscript. Overall, the authors have been very receptive to this reviewers suggestions. They include significantly functional data that was absent in the prior version. They also provide additional single cell analyses which improve the quality of the work. Congratulations to the authors on a job well done.

I have a few residual questions the authors should be able to answer without additional experiments.

In the authors human scRNA-seq data, I am wondering about how "mature" the macrophages are in the breast milk, and whether these are more like activated monocytes than like macrophages. I also wonder whether DCs have been adequately excluded.

1. Fig 6, the authors examine their own scRNA-seq of human milk MFs, and they show all 3 subsets have very low C1qa/C1qb - typically the highest expressing MF markers, but have increased FCN1 - a very good monocyte marker. They do map their 3 subsets to MF1, MF2 and MF3 is the MoMac-VERSE, but it would be important to map to monocytes (CD14+, cd16+) and also DC subsets. Can the authors show feature plots of the key MF, monocyte and DC genes, perhaps we can see transitions from monocyteMF?

2. In the reverse mapping of MoMac-VERSE to their milk MFs, DC2-DC3 is highly mapped, which confirms a better approach is needed here to define DCs / monocytes from the outset. The authors should readjust what these cells are called in humans.

3. There does not need to be symmetry between mouse and human (3 mouse MF subset = 3 human MF subsets). I suggest annotating human MF/DCs correctly from outset. Monocytes may be contributing to DC and MF repopulation continuously in human, and this would be important to know

Reviewer #2:

Remarks to the Author:

I thank the authors for addressing all of my concerns.

Author Rebuttal, first revision:

See Inserted PDF

Point-by-point Response 2

In the authors human scRNA-seq data, I am wondering about how “mature” the macrophages are in the breast milk, and whether these are more like activated monocytes than like macrophages. I also wonder whether DCs have been adequately excluded.

We thank the reviewer for this valid comment. As pointed out by the reviewer, mapping the human milk macrophages on the MoMac-Verse indeed revealed a small cluster within our original Mac 2 population that correlated to ‘DC2/DC3’ and one small cluster that mapped to ‘IL1B monocytes’ from the MoMac-Verse (previous Fig. 6h). We have now reclustered human milk macrophages and annotated DC2/DC3 and monocytes separately (new Fig. 6c-g, i and Ext. Fig. 7b-e).

1. Fig 6, the authors examine their own scRNA-seq of human milk MFs, and they show all 3 subsets have very low C1qa/C1qb - typically the highest expressing MF markers, but have increased FCN1 – a very good monocyte marker.

They do map their 3 subsets to MF1, MF2 and MF3 is the MoMac-VERSE, but it would be important to map to monocytes (CD14+, cd16+) and also DC subsets.

Can the authors show feature plots of the key MF, monocyte and DC genes, perhaps we can see transitions from monocyteMF?

We have reclustered Fig. 6c and mapped the DC2/DC3 and monocytes as separate clusters on the MoMac-Verse (Fig. 6g). We added expression of key macrophage, monocyte and DC genes to new Fig. 6d. With the new clusters, the highest expression of FCN1 is detected in monocytes, whereas C1QA and C1QB are higher in Macs 1-3 (new Fig. 6d).

Please find below also feature plots showing some of these genes (PbP 2 – Fig. 1). According to the feature plots, a clear monocyte to macrophage transition cannot be detected.

PbP 2 - Fig. 1. scRNA-seq. of human milk cells. Overlaid expression of different markers. Related to Figure 6c-d.

2. In the reverse mapping of MoMac-VERSE to their milk MFs, DC2-DC3 is highly mapped, which confirms a better approach is needed here to define DCs / monocytes from the outset. The authors should readjust what these cells are called in humans.

See the response above. We identified two extra clusters in the human milk in addition to the Macs 1-3: DC2/DC3 and monocytes; and we changed all the subsequent analysis accordingly (new Fig. 6c-g, i, new Ext. Fig. 7b-e).

3. There does not need to be symmetry between mouse and human (3 mouse MF subset = 3 human MF subsets). I suggest annotating human MF/DCs correctly from outset. Monocytes may be contributing to DC and MF repopulation continuously in human, and this would be important to know

As described above, we describe now two additional small populations of myeloid cells in human milk corresponding to DC2/DC3 and monocytes.

Decision Letter, second revision:

24th Mar 2023

Dear Dr. Greter,

Thank you for submitting your revised manuscript "Distinct lactation-associated macrophages exist in murine mammary tissue and human milk" (NI-A32115B). We are happy to inform you that if you revise your manuscript appropriately according to our editorial requirements, your manuscript should be publishable in Nature Immunology.

I will now pre-edit the current version of your paper. We will also perform detailed checks on your paper and will send you a checklist detailing our editorial and formatting requirements in about two-three weeks. Please do not upload the final materials and make any revisions until you receive this additional information from us.

In the meantime however, please deposit all omics and code data into public repositories so that the accession codes are readily available to be added in the revised manuscript. We cannot accept the paper without them. In addition, please check that your ORCID is linked to your Nature account, as this frequently causes delays at acceptance. Should you have any query or comments about ORCID, please do not hesitate to contact our editorial assistant at immunology@us.nature.com.

If you had not uploaded a Word file for the current version of the manuscript, we will need one before beginning the editing process; please email that to immunology@us.nature.com at your earliest convenience.

Thank you again for your interest in Nature Immunology. Please do not hesitate to contact me if you have any questions.

Sincerely,

Ioana Visan, Ph.D.
Senior Editor
Nature Immunology

Tel: 212-726-9207
Fax: 212-696-9752
www.nature.com/ni

Reviewer #1 (Remarks to the Author):

No further comments.

Final Decision Letter:

Dear Dr. Greter,

I am delighted to accept your manuscript entitled "Lactation-associated macrophages exist in murine mammary tissue and human milk" for publication in an upcoming issue of Nature Immunology.

Over the next few weeks, your paper will be copyedited to ensure that it conforms to Nature Immunology style. Once your paper is typeset, you will receive an email with a link to choose the appropriate publishing options for your paper and our Author Services team will be in touch regarding any additional information that may be required.

Please note that *Nature Immunology* is a Transformative Journal (TJ). Authors may publish their research with us through the traditional subscription access route or make their paper immediately open access through payment of an article-processing charge (APC). Authors will not be required to make a final decision about access to their article until it has been accepted. [Find out more about Transformative Journals](https://www.springernature.com/gp/open-research/transformative-journals).

Your paper will be published online soon after we receive your corrections and will appear in print in the next available issue. Content is published online weekly on Mondays and Thursdays, and the embargo is set at 16:00 London time (GMT)/11:00 am US Eastern time (EST) on the day of publication. Now is the time to inform your Public Relations or Press Office about your paper, as they might be interested in promoting its publication. This will allow them time to prepare an accurate and satisfactory press release. Include your manuscript tracking number (NI-A32115C) and the name of the journal, which they will need when they contact our office.

About one week before your paper is published online, we shall be distributing a press release to news organizations worldwide, which may very well include details of your work. We are happy for your institution or funding agency to prepare its own press release, but it must mention the embargo date and Nature Immunology. Our Press Office will contact you closer to the time of publication, but if you or your Press Office have any enquiries in the meantime, please contact press@nature.com.

Also, if you have any spectacular or outstanding figures or graphics associated with your manuscript - though not necessarily included with your submission - we'd be delighted to consider them as candidates for our cover. Simply send an electronic version (accompanied by a hard copy) to us with a possible cover caption enclosed.

Please note that we encourage the authors to self-archive their manuscript (the accepted version before copy editing) in their institutional repository, and in their funders' archives, six months after publication. Nature Portfolio recognizes the efforts of funding bodies to increase access of the research they fund, and strongly encourages authors to participate in such efforts. For information about our editorial policy, including license agreement and author copyright, please visit www.nature.com/ni/about/ed_policies/index.html

Sincerely,

Ioana Visan, Ph.D.
Senior Editor
Nature Immunology

Tel: 212-726-9207
Fax: 212-696-9752
www.nature.com/ni